**Observations of creep of polar firn at different temperatures**
Yuan Li[1,3], Kaitlin Keegan[2], Ian Baker[1]
[1]Thayer School of Engineering, Dartmouth College, Hanover, NH, 03755, USA
[2]Department of Geological Sciences & Engineering, University of Nevada, Reno, Reno, NV,
89557, USA
[3]X-Here (Future ice-based Hydrogen energy & resilient environments) Trek Laboratory
(Establishing), Howard Beach, NY, 11414, USA
*Correspondence to*: Ian Baker (ian.baker@dartmouth.edu)

**Abstract**

To improve our understanding of firn compaction and deformation processes, constant-load compressive creep tests were performed on specimens from a Summit, Greenland (72°35' N, 38°25' W) firn core that was extracted in June, 2017. Cylindrical specimens were tested at temperatures of –5°C, –18°C and –30°C from depths of 20 m, 40 m and 60 m at stresses of 0.21 MPa, 0.32 MPa and 0.43MPa, respectively. The microstructures were characterized before and after creep using both X-ray micro-computed tomography (micro-CT) and thin sections viewed between optical crossed polarizers. The results of these experiments comprise a novel data set on the creep of firn at three depths of a firn column at three different temperatures, providing useful calibration data for firn model development. Examining the resulting strain vs. time and strain vs. strain rate curves from the creep tests revealed the following notable features. First, the time exponent $k$ was found to be 0.34–0.69 during transient creep, which is greater than the 0.33 usually observed in fully-dense ice. Second, the strain rate minimum (SRmin) in secondary creep occurred at a greater strain from specimens with lower density and at higher temperatures. Third, tertiary creep occurred more easily for the lower-density specimens at greater effective stresses and higher temperatures, where strain softening is primarily due to recrystallization. Fourth, the SRmin is a function of the temperature for a given firn density. Lastly, we developed empirical equations for inferring the SRmin, as it is difficult to measure during creep at low temperatures. The creep behaviors of polar firn, being essentially different from full-density ice, imply that firn densification is an indispensable process within the snow-to-ice transition, particularly firn deformation at different temperatures connected to a changing climate.

**Keywords:** Firn densification; Creep; Activation energy; Cryospheric micro-CT; Temperature

## 1. Introduction

Understanding firn compaction and densification experimentally is critical for developing physics-based firn models that are necessary for many glaciological applications. For example, such models are essential for reconstructing ice-core paleoclimate records by simulating the lock-in depth of gases and the smoothing of climate signals (Schwander et al., 1997; Goujon et al., 2003). They are also crucial for interpreting ice-mass changes from satellite altimetry data, as they allow for the accurate correction of firn air content and surface elevation changes not related to underlying ice dynamics (Ligtenberg et al., 2011; Simonsen et al., 2013). However, the firn models used for these applications are empirical and are known to perform poorly outside of their calibration range (Lundin et al., 2017). Thus, a better understanding of firn compaction is necessary to refine firn models for these important glaciological applications. Laboratory compressive experiments on firn and ice improve our understanding of their respective flow laws and advance the development of firn models under a range of conditions. The rheology of polycrystalline ice, particularly its temperature-dependent creep deformation, is a cornerstone of glaciological modeling. Numerous studies have established a robust framework for understanding ice deformation, primarily through laboratory creep experiments (e.g. Glen, 1955; Weertman, 1983; Budd and Jacka, 1989; Durham et al., 2001; Goldsby and Kohlstedt, 2001; Petrenko and Whitworth, 1999). This body of work has confirmed that ice creep is strongly governed by temperature, typically described by an Arrhenius relationship with a well-constrained activation energy for grain-scale processes like dislocation glide and climb (e.g. Jacka, 1984; Hooke, 2005). In contrast, the mechanical behavior of firn, the intermediate porous material between snow and glacial ice, remains comparatively poorly characterized, especially with respect to temperature.

The experimental observations are interpreted by drawing parallels between firn deformation and the mechanical properties of its constituent material, polycrystalline ice. This connection is formalized through a poromechanics approach, where the behavior of the porous firn is derived from that of the ice skeleton using continuum mechanics and homogenization principles (Scapozza and Bartelt, 2003; Gagliardini and Meyssonnier, 2000; Coussy, 2004; Hutter and Johnk, 2004; Srivastava et al., 2010; Theile et al., 2011). While numerous studies have investigated ice deformation (e.g. Steinemann, 1954; Maeno and Ebinuma, 1983; Li et al., 1996; Jacka and Li, 2000; Song et al., 2006a, 2006b, 2008; Treverrow et al., 2012; Hammonds and Baker, 2016, 2018) and firn deformation (e.g. Landauer, 1958; Mellor, 1975; Salm, 1982; Ambach and Eisner, 1985; Meussen et al., 1999; Bartelt and von Moos, 2000; Theile et al., 2011; Li and Baker, 2021, 2022a), existing firn data are sparse and fragmented. A critical knowledge gap persists in the systematic experimental quantification of firn's mechanical response across a broad range of temperatures. Temperature is a first-order control on firn densification and deformation rates, yet most laboratory studies have been conducted at a limited number of isothermal conditions, often focused on a single density or at temperatures near the melting point (e.g. Mellor, 1975; Maeno and Ebinuma, 1983). Consequently, there is a pronounced lack of experimental data necessary to derive the systematic activation energy for the creep of firn over its full density spectrum. This parameter is not merely a scalar but is likely a function of density, microstructure, and the dominant deformation mechanism (compaction versus shear), transitioning from grain-boundary sliding in low-density firn to dislocation creep in high-density firn and ice (Hammonds and Baker, 2018; Li, 2022; Li and Baker, 2022a). The absence of comprehensive, temperature-variable creep data for firn across its density range renders it insufficient for constraining the

temperature-dependence terms in modern, physics-based firn models. Our work fills this gap via X-ray micro-computed tomography-analyzed mechanical examinations, e.g. a systematic series of constant-stress creep experiments on firn cores of varying density, conducted across a thermally controlled range from –30°C to –5°C. This allows for the direct determination whether the apparent activation energy is a function of density, thereby providing the essential experimental foundation needed to improve predictions of firn densification in ice-sheet and glacier models. Notably, the mechanical behavior of two-phase flow coupling the airflow with the ice matrix deformation has not yet been performed experimentally hitherto, even though the role of the microstructures of firn on airflow has been studied (Albert et al., 2000; Courville et al., 2010; Adolph and Albert, 2014). This difficulty is largely due to the limitations of the observation techniques of nondestructive visualization of the microstructures during snow and firn deformation. Thus, caution should be taken when extending the conclusions to ice sheet and glacier scales from sample laboratory experiments. Macroscopically, the creep of firn obeys a power-law dependence of the strain rate on the stress at constant stresses and temperature, similar to that of full-density ice (Li and Baker, 2022a). Note that both the diffusivity and permeability of the air in the pores (Albert et al., 2000; Courville et al., 2010; Adolph and Albert, 2014) impact heat conduction of the ice matrix, and hence the grain growth. This is tightly tied to the micro-mechanisms, e.g. grain-boundary and lattice diffusion of the ice crystals (Li and Baker, 2021), superplastic deformation and inter-particle sliding from dislocation motion in the ice necks (Bartelt and Von Moos, 2000), and likely rearrangement of the ice particles (Perutz and Seligman, 1939; Anderson and Benson, 1963; Ebinuma and Maeno, 1987).

Through experiments on isotropic ice samples subjected to uni-axial compaction at octahedral
stresses of 0.1–0.8 MPa and temperatures from –45ºC to –5ºC, Jacka and Li (2000) determined
the mechanisms involved in the empirical *power-law flow*, which was derived by Glen (1955) for
stresses ranging from 0.1–1 MPa at temperatures spanning from –13ºC to the melting-point. They
found that dynamic recrystallization predominated at higher temperatures and stresses, whereas
crystal rotation governed at lower temperatures and stresses. Later, Goldsby and Kohstedt (2001)
found that ice could exhibit *superplastic flow*, which depends inversely on the grain size,
particularly for fine-grained ice, while both dislocation creep and basal slip-limited creep were
unrelated to the grain size at stresses of 0.1 MPa or less over a wide range of temperatures.
Moreover, Baker and Gerberich (1979) reported that the apparent activation energy for creep for
polycrystalline ice, which was derived from tests at constant stress and temperatures ranging from
–40ºC to –5ºC, increased with increasing volume fraction of inclusions (bubbles, impurities, dust,
and air clathrate hydrates). Such inclusions governed the evolution of grain size related to thermal
activations. The activation energies for the creep of snow and ice have been determined by a
number of authors, and values ranging from 58.6–113 kJ mol$^{-1}$ were obtained under both uniaxial
and hydrostatic experiments for snow with a density of ~400 kg m$^{-3}$ at –13.6ºC to –3.6ºC
(Landauer, 1958); 44.8–74.5 kJ mol$^{-1}$ from snow with densities of 440–830 kg m$^{-3}$ at –34.5ºC to –
0.5ºC (Mellor and Smith, 1966); ~72.9 kJ mol$^{-1}$ for firn with a density of 320–650 kg m$^{-3}$ at the
South Pole (Gow, 1969); 69 $\pm$ 5 kJ mol$^{-1}$ for a mean snow density of 423 $\pm$ 8 kg m$^{-3}$ at –19ºC to
–11ºC (Scapozza and Bartelt, 2003); the 78 kJ mol$^{-1}$ from polycrystalline ice compression
deformation at a temperature of –10 ºC (Duval and Ashby, 1983); ~60 kJ mol$^{-1}$ for artificial and
natural ice at the South Pole (Pimienta and Duval, 1987); and 78 $\pm$ 4 kJ mol$^{-1}$ for monocrystal ice
at –20°C to –4.5°C and 75 ± 2 kJ mol$^{-1}$ for bicrystal ice at –15°C to –4.5°C (Homer and Glen,
1978). In summary, the flow law of polycrystalline ice and firn depends on the effects of
recrystallization, grain size, inclusions (Mellor and Testa, 1969; Vickers and Greenfield, 1968;
Barnes et al., 1971; Baker and Gerberich, 1979; Goodman et al., 1981), and the temperature.

With advanced observation techniques, the relevant microstructural parameters of snow and firn
have been characterized by a number of scientists (Arnaud et al., 1998; Coleou et al., 2001; Flin et
al., 2004; Wang and Baker, 2013; Wiese and Schneebeli, 2017; Li, 2022). Using X-ray
micro-computed tomography (micro-CT), Li and Baker (2022b) characterized metamorphism
from snow to depth hoar under opposing temperature gradients. Only rarely has work been
performed on the co-effects of temperature and stress on the densification of firn while
simultaneously visualizing the microstructural changes using a micro-CT. For example, Schleef et
al. (2014) reported that densification under varying conditions of overburden stress and
temperature from natural and laboratory-grown new snow showed a linear relationship between
density and the specific surface area (SSA). To this end, the aim of our present work is to
investigate the temperature dependence of the creep of polar firn and relate this to the change of
microstructure determined using micro-CT studies on firn obtained from Summit, Greenland in
2017. As is well known, temperature is a key parameter affecting the flow of firn and ice, and
plays a determined role in their deformation, especially for polythermal and temperate glaciers.
Due to the great difficulty of analyzing firn and ice deformation with the presence of liquid water,
this work focuses on the firn creep from the dry snow zone, i.e., areas without meltwater, at
different temperatures.

## 2. Samples and measurements

### 2.1 *Samples*

Three cylindrical samples (22 ± 0.5 mm diameter; 50 ± 0.5 mm high) were produced at each of three depths of 20 m, 40 m and 60 m from the same 2017 Summit, Greenland firn core that was studied in Li and Baker (2022a). Both the densities and porosities of these above samples are typical of values in the snow-to-ice transition zone as introduced in Section 1. It is important to note that the reduction in effective stress with increasing depth is evident in samples taken from these three specified depths (**Appendix A**). Before creep testing, one cylindrical firn samples from each depth was stored at a temperature of –5 ± 0.5ºC, –18 ± 0.5ºC, and –30 ± 0.5ºC for two days to achieve thermal equilibrium (Li and Baker, 2022a). It's also important to note that firn is a heterogeneous material that can have variations in layering, fabric, grain size, and impurity concentration across short distances. Thus, care was taken to extract the three replicate samples from the core at each depth as closely as possible to reduce the variability in their initial conditions.

### 2.2 Creep measurements

Three home-built creep jigs were placed in individual Styrofoam boxes in three different cold rooms that were held at temperatures of –5 ± 0.5ºC, –18 ± 0.5ºC and –30 ± 0.5ºC. Each creep jig consists of an aluminum base plate and three polished aluminum-guide rails passing through linear bearings that hold the upper aluminum loading plate (**Figure 1**). A linear voltage differential transducer (LVDT-Omega LD-320: resolution of 0.025%; linearity error of ± 0.15%

of full-scale output), parallel to the three aluminum-guide rails, was located adjacent to the center
of the upper plate, and fixed firmly using a screw through the plate (**Figure 1**) for measuring the
displacement during a test. The displacement was logged every 5 seconds using a Grant SQ2010
datalogger (accuracy of 0.1%). Temperatures were logged at 300-second time intervals over the
entire test period, using a k-type thermocouple (Omega RDXL4SD thermistor: resolution of 0.1ºC)
that was mounted inside each box. In this work, specimens were tested at temperatures of –5 ±
0.2ºC, –18 ± 0.2ºC and –30 ± 0.2ºC from depths of 20 m (applied stress 0.21 MPa), 40 m (0.32
MPa) and 60 m (0.43 MPa). There are smaller error bars for the temperature of the specimens
than the room temperature because the creep jigs were in insulated Styrofoam boxes. The stresses
were chosen based on experience from previous tests (Li and Baker, 2022a) in order to give
measurable creep rates in a reasonable time.

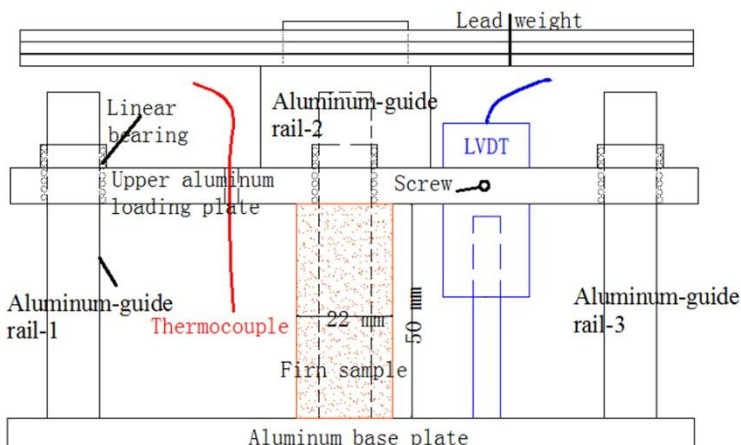


**Figure 1:** Schematic illustrating the home-built compressive creep jigs. More details can be found
in (Li & Baker, 2022a).


2.3 *X-ray micro-computed tomography (micro-CT)*
Each specimen at each depth and temperature combination was scanned using a Skyscan 1172
micro-CT, before and after creep testing. Each micro-CT scan lasted ~2 h. The cubic Volume of
Interest (VOI, a side length of 8 mm) was taken from near the center of the firn specimen as
conducted in Li and Baker (2022a). The microstructural parameters obtained from the micro-CT
data are the SSA, the mean structure thickness of the ice matrix (S.Th), the area-equivalent circle
diameter of the pores (ECDa), the total porosity (TP), the closed porosity (CP), and the structure
model index (SMI). The SSA ($mm^{-1}$) is the ratio of the ice surface area to total firn volume (ice
plus air) in a VOI analytical element, and is calculated using the hexahedral marching cubes
algorithm via CTAn software (Wang and Baker, 2013). It characterizes the thickness and
complexity of the firn microstructure. Changes in SSA indicate a change in free energy of the ice
surfaces, the decrease of which represents the occurrence of sintering-pressure. The S.Th (mm) is
the mean structure thickness of an ice matrix (Hildebrand and Ruegsegger, 1997), which
represents the characteristic size of an ice particle in the firn, where the ice particle consists of one
or many crystals or grains. It is measured based on the largest sphere diameter that encloses a
point in the ice matrix and is completely bounded within solid surfaces. The ECDa (mm) is the
diameter of a circle having the same area as the average for all pores in the VOI from the 2-D
binary images, indicative of the characteristic size for the void space (Adolph and Albert, 2014).
The TP (%) is the ratio of the pore volume, including both open and closed pores, to the total VOI.
The CP (%) is the ratio of the volume of the closed pores to the total volume of solid plus closed
pores volume in a VOI, while the open porosity (%) is the ratio of the volume of the open pores to
the total VOI. The SMI is calculated based on the dilation of a 3-D voxel model (Hildebrand and
Ruegsegger, 1997) $\mathrm{SMI} = 6\left(S' \times V\right)\big/S^2$, where $S'$ is the change in the surface area due to
dilation, and $V$ and $S$ are the object volume and surface area, respectively. It indicates the
prevalent ice curvature, negative values of which represent a concave surface, e.g. the hollow air
structure surrounded by an ice matrix. The more negative the SMI value, the more spherical the
pore. Notably, the micro-CT-derived density of each specimen agrees well with the bulk density
measured using the mass-volume approach (Li and Baker, 2021).

*2.4 Thin section preparation and imaging*
Thin sections for optical photographs before and after creep testing were cut from bulk specimens,
one side of which was first smoothed with a microtome. This side was then frozen onto a glass
plate ($100 \times 60 \times 2$ mm) by dropping supercooled gas-free water along its edges. Its thickness
was reduced to ~2 mm by a band saw, and finally thinned further to a uniform thickness of ~0.5
mm using a microtome. Images were captured using a digital camera after each thin section was
placed on a light table between a pair of crossed polarizing sheets.

**3 Results and discussion**
3.1 *Microstructures before creep*
Increasing firn density with increasing depth from either of the –5°C, –18°C, and –30°C
specimens can be readily recognized by visual inspection of the micro-CT 3-D reconstructions of
the firn microstructure (**Figure 2**). Correspondingly, the microstructural parameters, with the
exception of the CP, changed monotonically with increasing depth at each temperature, e.g. the –
30°C samples increased in density from $591 \pm 1.4$ kg m$^{-3}$, to $683 \pm 4.2$ kg m$^{-3}$, to $782 \pm 1.5$ kg m$^{-3}$,
decreased in SSA from $4.64 \pm 0.04$ mm$^{-1}$, to $3.3 \pm 0.06$ mm$^{-1}$, to $2.39 \pm 0.01$ mm$^{-1}$, and decreased
in TP from $35.6 \pm 0.05\%$, $25.6 \pm 0.4\%$, to $14.8 \pm 0.2\%$ at 20, 40, and 60 m, respectively (**Table 1)**.
These above changes are similar to those previously observed in this firn core (Li and Baker,
2022a), implying that the sintering-pressure mechanism plays a crucial role in the densification of
polar firn due to the increasing overburden of snow and firn with increasing depth. However, the
microstructures of the samples from the three temperatures at each depth show little variability
and do not monotonically change with temperature, e.g. at 20 m depth the $-5$°C, $-18$°C, and $-$
30°C samples having densities of $589 \pm 1.3$ kg m$^{-3}$, $615 \pm 2.5$ kg m$^{-3}$, and $591 \pm 1.4$ kg m$^{-3}$, and

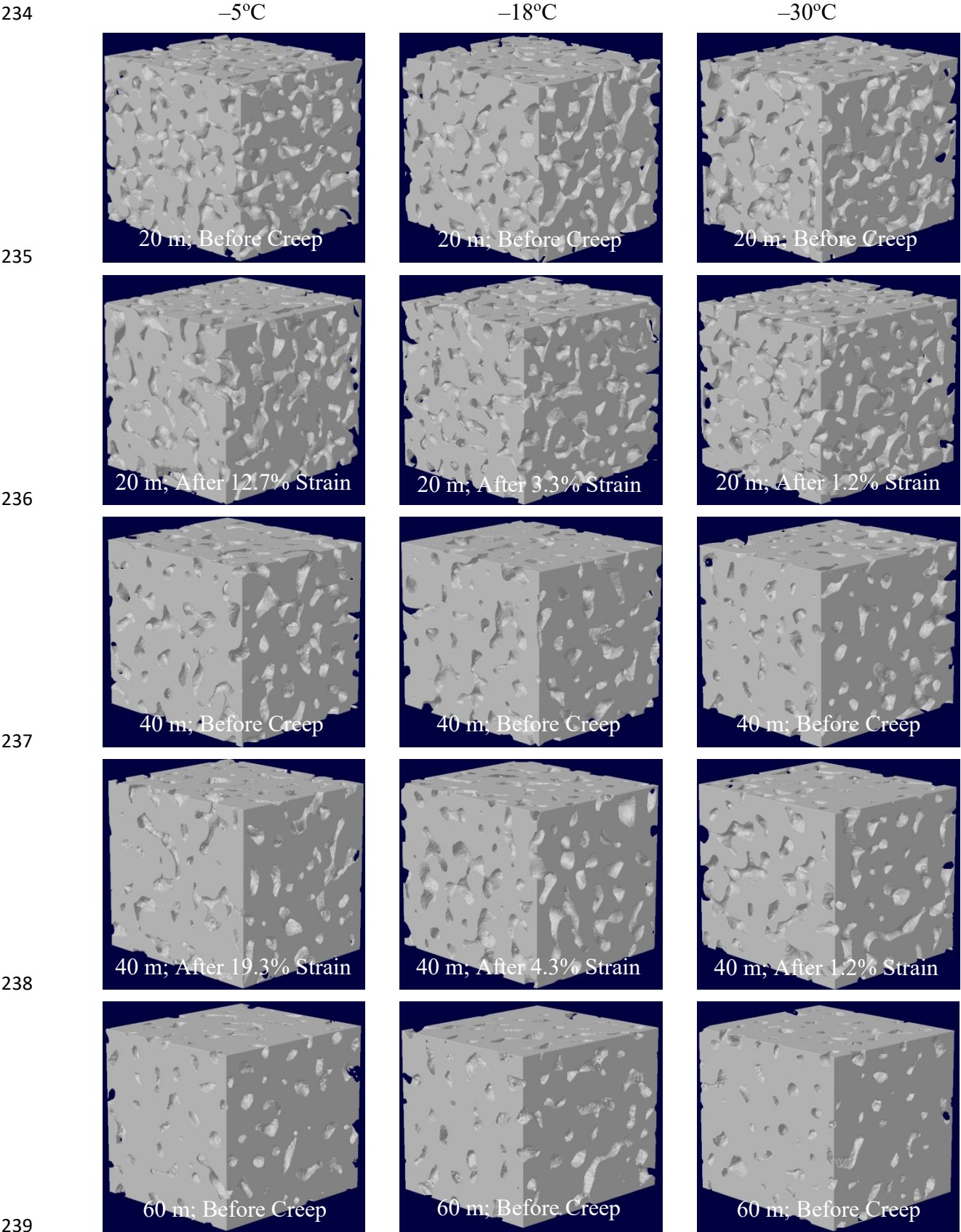

−5ºC       −18ºC       −30ºC






20 m; Before Creep

20 m; After 12.7% Strain    20 m; After 3.3% Strain    20 m; After 1.2% Strain

40 m; Before Creep

40 m; After 19.3% Strain    40 m; After 4.3% Strain    40 m; After 1.2% Strain

60 m; Before Creep

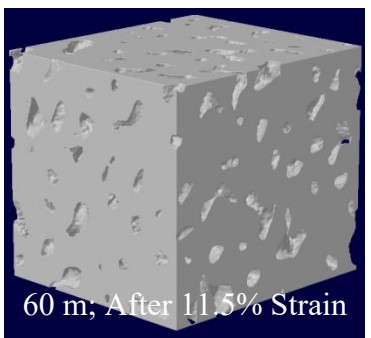 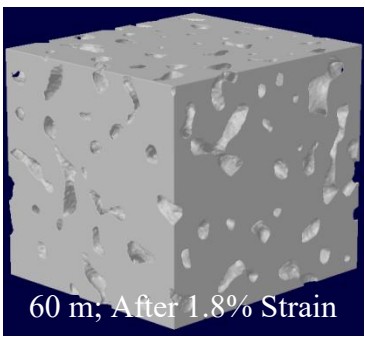 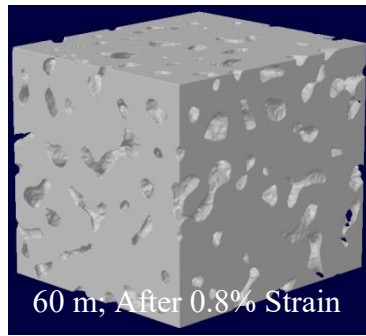


**Figure 2:** Micro-CT 3-D reconstructions (the side length of each cubic volume of interest is 8 mm)
of specimens before and after creep testing at the depths and temperatures shown. Grey voxels
represent ice in the firn structure.


Table 1. Microstructural parameters derived from Micro-CT for samples at –5ºC, –18ºC, and –
30ºC from depths of 20 m, 40 m, and 60 m before creep.

| | | | | 20m | | | | |
|---|---|---|---|---|---|---|---|---|
| T | Density | SSA | S.Th | TP | CP | | | ECDa |
| ºC | kg m⁻³ | mm⁻¹ | mm | % | % | | SMI | mm |
| –5 | 589±1.3 | 4.74±0.03 | 0.732±0.001 | 35.9±0.08 | 0.03±0.002 | -0.31±0.04 | | 1.07±0.005 |
| –18 | 615±2.5 | 4.51±0.04 | 0.758±0.001 | 33.1±0.2 | 0.01±0.001 | -0.57±0.01 | | 0.995±0.013 |
| –30 | 591±1.4 | 4.64±0.04 | 0.747±0.004 | 35.6±0.05 | 0.02±0.001 | -0.27±0.05 | | 1.09±0.004 |


| | | | | 40m | | | | |
|---|---|---|---|---|---|---|---|---|
| T | Density | SSA | S.Th | TP | CP | | | ECDa |
| ºC | kg m⁻³ | mm⁻¹ | mm | % | % | | SMI | mm |
| –5 | 685±1.4 | 3.26±0.04 | 0.95±0.004 | 25.5±0.1 | 0.015±0.001 | -1.85±0.11 | | 0.857 ±0.005 |
| –18 | 669±0.6 | 3.41±0.04 | 0.914±0.003 | 27.1±0.03 | 0.01±0.001 | -1.69±0.09 | | 0.892±0.002 |
| –30 | 683±4.2 | 3.30±0.06 | 0.94±0.01 | 25.6±0.4 | 0.021±0.005 | -1.87±0.13 | | 0.86±0.01 |


| | | | | 60m | | | | |
|---|---|---|---|---|---|---|---|---|
| T | Density | SSA | S.Th | TP | CP | | | ECDa |
| ºC | kg m⁻³ | mm⁻¹ | mm | % | % | | SMI | mm |
| –5 | 790±1.0 | 2.34±0.03 | 1.1±0.003 | 14.0±0.1 | 0.11±0.01 | -4.81±0.22 | | 0.594±0.003 |
| –18 | 780±0.5 | 2.37±0.03 | 1.08±0.01 | 15.2±0.06 | 0.014±0.007 | -4.8±0.2 | | 0.632±0.001 |
| –30 | 782±1.5 | 2.39±0.01 | 1.076±0.0003 | 14.8±0.2 | 0.02±0.001 | -4.92±0.14 | | 0.639±0.002 |

Note: SSA is the specific surface area, S.Th is the structure thickness, TP is the total porosity, CP
is the closed porosity, SMI is the structure model index, and ECDa is the area-equivalent circle
diameter.


SSAs of $4.74 \pm 0.03$ mm$^{-1}$, $4.51 \pm 0.04$ mm$^{-1}$, and $4.64 \pm 0.04$ mm$^{-1}$, respectively (**Figures 2–3**;
**Table 1)**. Here, despite the $-18^{\circ}$C specimen having a higher density than the two others at $-5^{\circ}$C
and $-30^{\circ}$C, is not possible to conclude that the sintering of firn is not directly related to the
temperature. This is likely because a thermal equilibration period of two days in the absence of
compression is too short to sufficiently exert the influence of temperature on firn sintering. The
microstructural differences seen in these specimens more likely arose from the initial samples
themselves, which were anisotropic and heterogeneous even if taken from the same depth,
attributed to firn pre-deformation and partial annealing before experiments (Li and Baker, 2022a).

3.2 *Microstructures after creep*
The microstructural evolution is characterized by the microstructural parameters shown in **Figure**
**3**. The largest changes occurred in the $-5^{\circ}$C specimens due to the higher temperature, i.e., the
density, S.Th, and CP increased, while the ECDa, TP, SSA, and SMI decreased, indicative of
consolidation of the firn after creep. It is important to note that for the 60 m sample tested at $-5^{\circ}$C,
there was no change in density, i.e., $790.2 \pm 1$ kg m$^{-3}$ before creep vs. $790.7 \pm 0.9$ kg m$^{-3}$ after
creep, or TP, i.e., $14.0 \pm 0.1\%$ before creep vs. $13.9 \pm 0.1\%$ after creep. This lack of
microstructural change is due to the high initial density, which was close to the firn pore close-off
density of $\sim$830 kg m$^{-3}$. Thus, the creep of this sample may involve a transition from firn to
bubbly ice, as is also indicated by the increase in CP, which would have made it difficult to
compress further. Intriguingly, some of the changes in microstructure observed in the micro-CT
3-D reconstructions from the specimens before and after creep, e.g. the distribution of ice-space,
are indistinguishable in **Figure 2**. This is presumably due to the relatively large initial particle size,
or from radial dilation exceeding the axial compression because of the small strains that occurred
at relatively low temperatures.

One exception to the expected microstructural change after creep was the decrease of CP, which
was likely due to the measurement uncertainty of the micro-CT (Burr et al., 2018), or radial
expansion of the specimen during creep. Another exception was the decrease in density after
creep for the –18°C specimen at 20 m and the –30°C specimen at 60 m, which arose due to a
de-densification effect produced by temperature gradient metamorphism, as confirmed by the
increase of both TP and S.Th (Li and Baker, 2022b). The thermal gradient appears to be
associated with a fluctuation of 0.2°C around the test temperature, similar to temperature cycling
occurred within firn (Mellor and Testa, 1969; Weertman, 1985), which stems from the
thermometer's inherent accuracy as noted in Section 2.2 (–5 ± 0.2°C, –18 ± 0.2°C and –30 ±
0.2°C). In the relatively simple deformation found at ice-sheet dome sites, such as Summit, there
is no mechanism to decrease density during compression. At sites closer to the ice sheet margins,
cracking due to extension of the ice may cause a localized decrease in density. The rate of firn
densification should decrease with increasing depth at a given temperature, due to the decrease of
effective stress with increasing depth (**Appendix A**). As a matter of fact, the density of the –5°C
samples after creep increased by 32 kg m$^{-3}$, 44 kg m$^{-3}$, and 0.5 kg m$^{-3}$ for the 20 m, 40

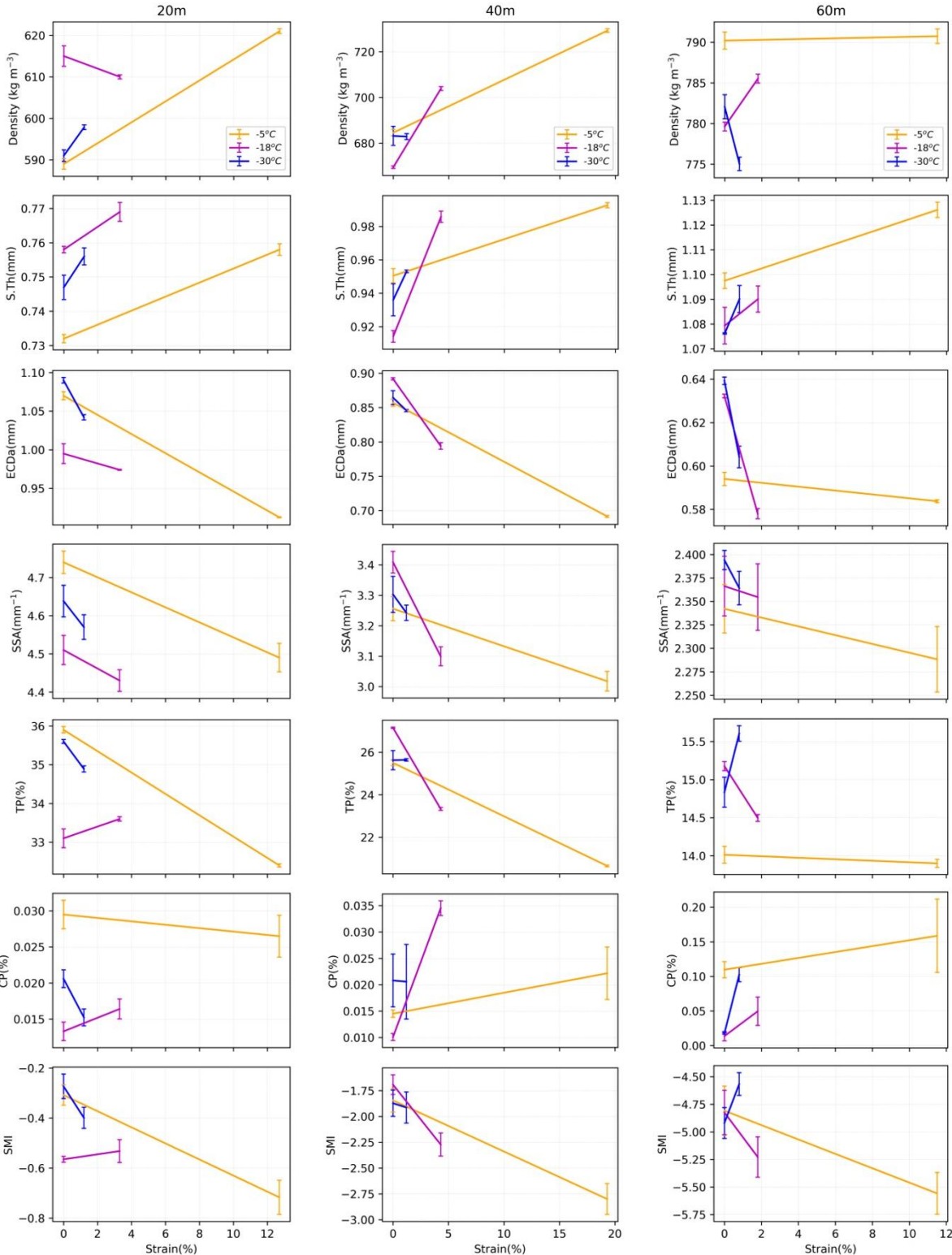

**Figure 3:** Density, structure thickness (S.Th), area-equivalent circle diameter (ECDa), specific

surface area (SSA), total porosity (TP), closed porosity (CP), and structure model index (SMI) of

the firn samples before and after creep at three temperatures (orange, magenta, and blue lines)
from depths of 20 m, 40 m and 60 m. Error bars indicate the variation of each microstructural
parameter as derived from three different VOIs of the same sample.


m, and 60 m samples, respectively. The 44 kg m$^{-3}$ unexpectedly outnumbers the 32 kg m$^{-3}$,
implying that the densification of firn is also affected by other undetermined factors, e.g. the
effect of inclusions, in addition to the stress and temperature.

Another way to investigate microstructure changes before and after creep tests is to compare their
grain sizes using thin sections. As an example, **Figure 4** shows optical micrographs of thin
sections made from the –5°C sample at 40 m before and after creep to a strain of 19.3%, where
the significant reduction in grain size from 0.8 ± 0.67 mm to 0.5 ± 0.32 mm implies the
occurrence of recrystallization during testing. However, it is also unclear at what strain
recrystallization








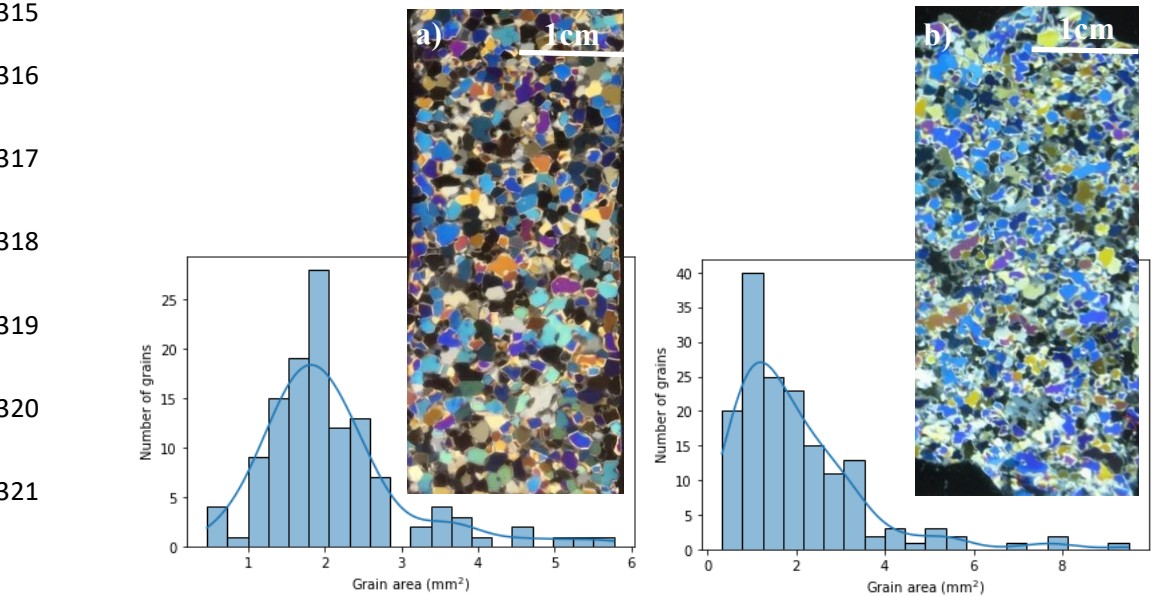


**Figure 4:** Optical micrographs of thin sections, and the distribution of grain sizes for the 40 m sample
at –5°C (a) before and (b) after creep (19.3% strain).


was initiated in each test, as noted in Li and Baker (2022a). Recrystallization occurs frequently at
a temperature higher than the homologous temperatures of 0.9 $T_m$. However, no evidence was
found for recrystallization after testing at the relatively cold –18°C and –30°C conditions,
probably due to the small creep strains at these relatively low temperatures. The creep
mechanisms for these samples, and whether the mechanisms were different at different
temperatures, could not be determined from the micro-CT-derived microstructural observations
alone, because the micro-CT can only capture the microstructure before and after creep. Instead,
plots of both strain vs. time and strain rate vs. strain can be used to elucidate the onset of
recrystallization during creep (Sections 3.3 and 3.4; Ogunmolasuyi, et al., 2023).

*3.3 Relationship between strain and time*
**Figure 5** shows the strain vs. time creep curves. The specimens at –5°C at 20 m and –18°C at 20
m, 40 m, and 60 m, show decelerating transient creep and quasi-viscous steady-state creep, while
the specimens at –5°C at 40 m and 60 m show transient, secondary, and accelerating tertiary creep.
Note that the curves from the –30°C specimens are not easily interpreted due to a large amount of
noise arising from both the insufficient resolution of a linear voltage differential transducer (Li
and Baker, 2022a) and the very small strains. The transient creep stage may be caused by strain
hardening that occurs from the yield point to the ultimate strength (Glen, 1955; Jacka, 1984). The
plastic deformation is accommodated by an increase in dislocation density through dislocation
multiplication or the formation of new dislocations (Frost and Ashby, 1982; Duval et al., 1983;
Ashby and Duval, 1985), which leads to an increase of the firn strength as the dislocations
become pinned or tangled, and thus more difficult to move. The initial decrease of creep rate may
also be related to the rearrangement of dislocations into a more stable pattern through a dragging
mechanism (Weertman, 1983) for the –5°C specimens. The tertiary creep stage may be associated
with strain softening deriving either from the thermally-activated processes at the high
homologous temperature approaching the melting point of ice, or from recrystallization (Li and
Baker, 2022a). Clearly, the creep rate of firn is sensitive to temperature under constant stress at a
given depth, *viz.*, the creep rate increases with increasing temperature (**Figure 5**). Incidentally,
there is no evidence of the onset of recrystallization in the creep curves themselves despite the
thin-section observation that –5°C specimens clearly underwent recrystallization during creep
(Section 3.2).

A modified Andrade-like equation $\varepsilon = \beta t^k + \varepsilon_0$ in Li and Baker (2022a) was used to describe the
transient creep behavior of the firn, in which the primary creep was well represented in black
dashed lines on the creep curves in **Figure 5**. The time exponent $k$, derived from the above
equation, ranges from 0.34–0.69: the data for the –30°C specimens are excluded since the noise in
the results makes them uninterpretable. These $k$ values are also smaller than those from
monocrystalline and bicrystalline ice: 1.9 ± 0.5, 1.5 ± 0.2, and 1.3 ± 0.4 (Li and Baker, 2022a and
references therein). We also note that the $k$ values from the specimens at –5°C from 20–60 m

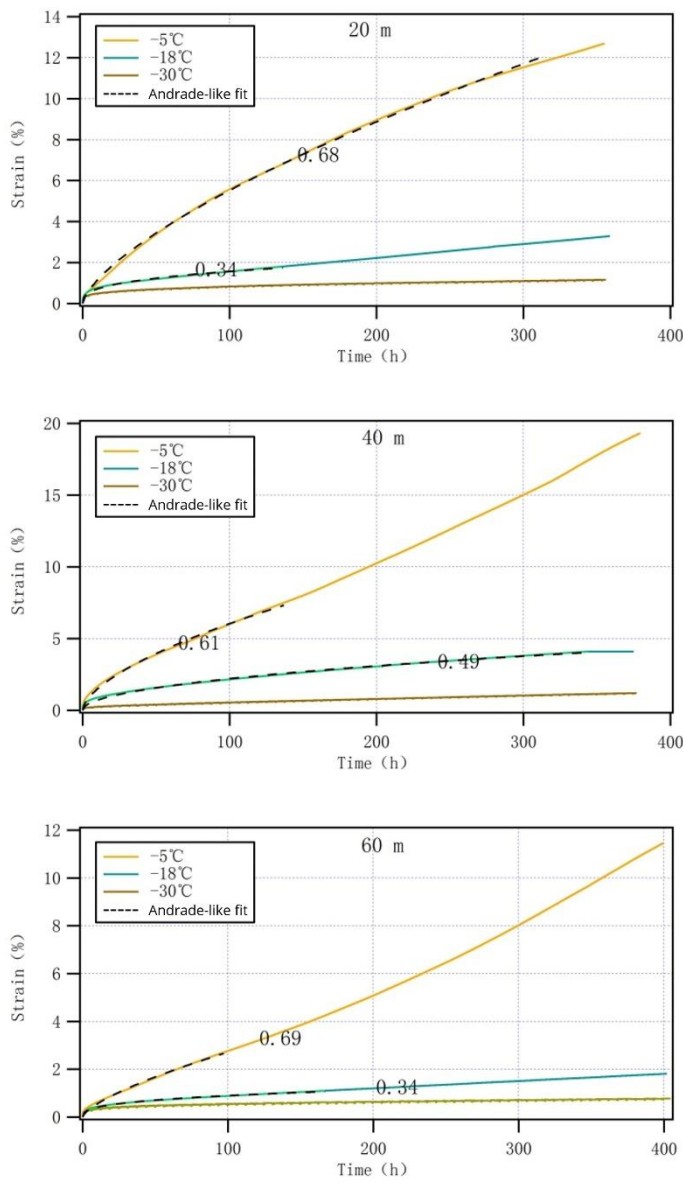

**Figure 5:** Strain vs. time for firn specimens at –5°C (yellow lines), –18°C (blue lines), and –30°C (brown lines), from depths of 20 m (applied stress 0.21 MPa), 40 m (0.32 MPa) and 60 m (0.43 MPa). The black dashed curves represent fits to a modified Andrade-like equation with the time exponents indicated on the curves, if any. Note: The *y*-axis limits vary across the subfigures.

(0.68, 0.61, and 0.69), and at –18°C from 40 m (0.49) are greater than 0.33, while the *k* value from the –18°C specimens at 20 m (0.34) and 60 m (0.34) are close to 0.33 that is usually

obtained for full-density polycrystal ice. Interestingly, an evident relationship between the density
of firn and the $k$ values, regardless of the effect of stress (Li and Baker, 2022a) and temperature,
remains unknown. A greater $k$ value signifies faster deformation. The $k$ values derived for firn are
generally higher than those for polycrystalline ice, implying that the higher firn deformation rates
compared to those of ice ($k = \sim 0.33$; Cuffey and Paterson, 2010, and references therein) are likely
related to the fewer grain-boundary constraints with more void space in firn (Li and Baker, 2022a;
Li, 2023b). Clearly, the above $k$ values, which increased with increasing temperature (**Figure 5**),
indicate that deformation is easier because of the lower viscosity at the higher temperature. Thus,
$k$ seems to be a state variable with respect to temperature. In addition, $k$ values greater than 0.33
may be related to the decrease of viscosity of the firn specimens (Freitag et al., 2002; Fujita et al.,
2014). $k$ values lower than 0.33 observed under constant load and temperature occurred at
relatively low effective stresses (Li and Baker, 2022a). The identified trend of steadily declining $k$
values across the temperature range of –5°C to –18°C, however, represents a significant gap in
our current understanding, necessitating a dedicated investigation into the microstructural or
metamorphic causes. Alternatively, the enhanced cohesion strength in the firn, which resulted
from both the ice matrix with higher purity and the stronger bond connection of inter-grains,
increases the viscosity of test samples and lowers the $k$ value to less than 0.33.

3.4 *Relationship of strain rate to strain*
**Figure 6** shows log strain rate vs. strain plots from all the –5ºC and –18ºC specimens; the –30ºC
samples are excluded due to noise. The evolution of the strain rate is characterized more clearly in
**Figure 6** than in **Figure 5**. Clearly, the strain rate is also a state variable of temperature, where the
strain rate increases with increasing temperature for a given strain at a given depth (**Figure 6**;
**Table 2**). The strain rate minimum at the secondary creep stage (SRmin) and the strain at the


Table 2. Observed and inferred strain rate minima and strains observed at the strain rate minima.

| 20 m | SRmin $s^{-1}$ | PC1-SRmin $s^{-1}$ | PC2-SRmin $s^{-1}$ | PC3-SRmin $s^{-1}$ | Strain % |
|---|---|---|---|---|---|
| –5°C | $5.53 \times 10^{-6}$ | $\mathit{5.53 \times 10^{-6}}$ | $\mathit{1.68 \times 10^{-6}}$ | $\mathit{2.56 \times 10^{-7}}$ | 11.8 |
| –18°C | $1.36 \times 10^{-6}$ | $\mathit{1.36 \times 10^{-6}}$ | $\mathit{2.29 \times 10^{-7}}$ | $\mathit{2.45 \times 10^{-8}}$ | 1.81–2.9 |
| –30°C(U) | – | $7.14 \times 10^{-7}$ | $2.17 \times 10^{-7}$ | $3.3 \times 10^{-8}$ | – |
| –30°C(L) | – | $3.16 \times 10^{-8}$ | $9.6 \times 10^{-9}$ | $1.46 \times 10^{-9}$ | – |


| 40 m | SRmin $s^{-1}$ | PC1-SRmin $s^{-1}$ | PC2-SRmin $s^{-1}$ | PC3-SRmin $s^{-1}$ | Strain % |
|---|---|---|---|---|---|
| –5°C | $1.03 \times 10^{-5}$ | $\mathit{3.39 \times 10^{-5}}$ | $\mathbf{\mathit{1.03 \times 10^{-5}}}$ | $\mathit{1.57 \times 10^{-6}}$ | 7.5 |
| –18°C | $1.4 \times 10^{-6}$ | $\mathit{8.32 \times 10^{-6}}$ | $\mathbf{\mathit{1.40 \times 10^{-6}}}$ | $\mathit{1.5 \times 10^{-7}}$ | 4.1 |
| –30°C(U) | – | $4.37 \times 10^{-6}$ | $1.33 \times 10^{-6}$ | $2.03 \times 10^{-7}$ | – |
| –30°C(L) | – | $1.94 \times 10^{-7}$ | $5.88 \times 10^{-8}$ | $8.97 \times 10^{-9}$ | – |


| 60 m | SRmin $s^{-1}$ | PC1-SRmin $s^{-1}$ | PC2-SRmin $s^{-1}$ | PC3-SRmin $s^{-1}$ | Strain % |
|---|---|---|---|---|---|
| –5°C | $5.59 \times 10^{-6}$ | $\mathit{1.21 \times 10^{-4}}$ | $\mathit{3.67 \times 10^{-5}}$ | $\mathbf{\mathit{5.59 \times 10^{-6}}}$ | 2.7 |
| –18°C | $5.33 \times 10^{-7}$ | $\mathit{2.96 \times 10^{-5}}$ | $\mathit{4.99 \times 10^{-6}}$ | $\mathbf{\mathit{5.33 \times 10^{-7}}}$ | 1.1–1.8 |
| –30°C(U) | – | $1.56 \times 10^{-5}$ | $4.74 \times 10^{-6}$ | $7.21 \times 10^{-7}$ | – |
| –30°C(L) | – | $6.91 \times 10^{-7}$ | $2.1 \times 10^{-7}$ | $3.19 \times 10^{-8}$ | – |

The SRmin without the prefix is the observed values during creep, while the SRmin with a prefix
is the inferred values. Note that PC-SRmin is the abbreviation of the post-calibration SRmin, and
that –30°C(U) and –30°C(L) indicate the upper and lower bound from the –30°C samples from
44.8 kJ mol$^{-1}$ and 113 kJ mol$^{-1}$, respectively. PC1-SRmin, PC2-SRmin, and PC3-SRmin are
described in **Appendix B**. The symbol – indicates the unavailable values of SRmin and the strain
value at the SRmin observed during creep. For the italics highlighted, see **Appendix B**.


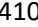




**Figure 6:** Log strain rate vs. strain from the firn specimens at temperatures of -5°C and -18°C

from depths of 20 m (applied stress 0.21 MPa), 40 m (0.32MPa) and 60 m (0.43MPa). Samples

from -30°C are not shown due to the very large noise. The blue lines represent discrete strain rates,

which are calculated by extracting the strain data hourly, while the orange lines represent a

moving average of 15 moving windows with respect to the strain.



SRmin for all the –5ºC and –18ºC specimens are shown in **Figure 6** and **Table 2**. The SRmin was
reached at a strain of 11.8%, 7.5% and 2.7% for the –5ºC specimens from depths of 20 m, 40 m,
and 60 m, respectively, consistent with strains at the SRmin decreasing with increasing depth at a
given temperature in **Figure 8** and **Table 4** in Li and Baker (2022a). For the –18 ºC specimens,
the SRmin occurred over a range of strains from 1.81–2.9% at 20 m, at a fixed strain of 4.1% at
40 m, and at a strain oscillating between 1.1 and 1.8% at 60 m. These values of strain at different
SRmin values are different from those usually observed at strains of 0.5–3% for fully-dense ice
(Cuffey and Paterson, 2010, and references therein), implying different mechanical behavior
between firn and pure ice (Duval, 1981; Mellor and Cole, 1983; Jacka, 1984; Li et al., 1996;
Jacka and Li, 2000; Song et al., 2005, 2008; Cuffey and Paterson, 2010). Overall, the strain at the
SRmin is greater with lower density and higher temperature, e.g. 11.8% strain from the –5ºC
specimens at 20 m, and 4.1% strain from the –18ºC specimens at 40 m. This is likely due to the
effect of strain hardening on density and temperature (Li, 2023b). Additionally, tertiary creep is
observed during both quasi-steady state deformation, particularly in the –5°C specimens at depths
of 40 m and 60 m, and in the ascending stage, as seen in the –5°C and –18°C specimens at 20 m,
along with the –18°C specimen at 40 m. This mechanical behavior is facilitated by lower firn
density, increased effective stress, and elevated creep temperatures. For instance, in the –5°C
specimens at 20 m, strain softening primarily results from recrystallization (Duval, 1981; Jacka,
1984; Jacka and Li, 2000; Song et al., 2005; Faria et al., 2014). Also, the activation of easy slip
systems contributes to this process (Jonas and Muller, 1969; Duval and Montagnat, 2002; Alley et
al., 2005; Horhold et al., 2012; Fujita et al., 2014; Eichler et al., 2017). It is noteworthy that Jacka
and Li (1994) observed that steady-state tertiary ice creep, which is marked by stable grain size, is
influenced more by applied stresses than by temperature. This finding suggests that there exists a
balance between the activation energies required for grain growth and subdivision at a specific
temperature.

3.5 *Apparent activation energy for creep*
Experimental observations of the SRmin are limited, as they only occurred for the –5°C and at –
18°C specimens at each depth (**Table 2**). It is hard to achieve the SRmin for all firn specimens in
laboratory environments (Landauer, 1958), especially under low temperatures and stresses such as
those from the –30°C specimens in this work. To this end, we offer the various possibilities of the
SRmin using the evidence we have. The value of the apparent activation energy of creep, $Q_c$ (kJ
mol⁻¹), is equal to the slope of a line fitted $\ln \dot{\varepsilon}$ versus $1/T$ as did in Goldsby & Kohlstedt (1997;
2001), using the Arrhenius relation $\dot{\varepsilon} = B\sigma^n \exp(-\frac{Q_c}{RT})$, where $\dot{\varepsilon}$ (s⁻¹) is the strain rate, $B$ (s⁻¹
Pa⁻ⁿ) is the material parameter, $\sigma$ (MPa) is the applied stress, $n$ is the creep (stress) exponent, $R$
(8.314 J mol⁻¹ K⁻¹) is the gas constant, and $T$ (K) is Kelvin temperature. First, the estimation of
$Q_c$ is based on *only* two SRmin values from the –5°C and –18°C samples at each depth (**Table 2**).
Glen-King's model $\dot{\varepsilon} = A\exp(-Q_c/RT) = B\sigma^n \exp(-Q_c/RT)$ treats the pre-factor $A$, material
parameter $B$, and stress exponent $n$ as constants (Glen, 1955; Goldsby and Kohlstedt, 2001). This
simplification is valid by using the unifying concept of normalized effective stress. The effective
stress captures the complex multi-physical behavior of the two-phase ice-air system, accounting
for: 1) The incompressibility of individual ice grains versus the compressibility of the porous ice
skeleton, 2) The coupled flow of ice and air; and 3) The interplay between different strain
components (axial, radial, volumetric, and true). This framework is grounded in the principles of
poromechanics, originally developed for soils and later applied to snow and ice (Gubler, 1978;
Hansen and Brown, 1988; Mahajan and Brown, 1993; Chen and Chen, 1997; Lade and deBoer,
1997; Ehlers, 2002; Khalili et al., 2004; Gray and Schrefler, 2007; daSilva et al., 2008; Nuth and
Laloui, 2008). The variability in density for the samples from 20-m depth on the mechanical
behavior are negligible due to a small difference (up to ~4%), between samples, which falls
within an acceptable error range in previous studies. This is likely related to multiple factors,
including the intrinsic properties of the samples, e.g. inclusions (impurities, dust, bubbles,
clathrate hydrates), the effects of deformation and partial annealing of firn due to stress
distribution and temperature changes during drilling, extraction, transportation, or storage, and the
fact that the samples are taken from adjacent parts of the core, and might capture heterogeneous
density layers, as well as potential measurement errors associated with the equipment used. The
$Q_c$ values from the 20 m, 40 m, and 60 m specimens were calculated to be 61.4 kJ mol$^{-1}$, 87.3 kJ
mol$^{-1}$, and 102.8 kJ mol$^{-1}$, respectively (**Figure 7**). Based on the three SRmin from the –5$^{\circ}$C and –
18$^{\circ}$C samples at 60 m in this work, and from –10$^{\circ}$C samples at 60 m in Li and Baker (2022a), a
$Q_c$ value for the 60 m specimen was calculated to be 100.7 kJ mol$^{-1}$. To see whether or not these
above $Q_c$ values are reliable, we estimated the activation energy of grain-boundary
diffusion/viscosity, $Q_{gbd}$ (kJ mol$^{-1}$), using the relation $K = \left( D_t^2 - D_0^2 \right) \big/ t = k \exp(-Q_{gbd}/RT)$,
in an alternative form of $Q_{gbd} = -R \left[ \partial \ln K \big/ \partial (1/T) \right]$, where $K$ is the observed rate of grain
growth (mm$^2$ a$^{-1}$), $D_0^2$ and $D_t^2$ are the measured mean grain area (mm$^2$) in a firn sample at the
onset of the creep ($t = 0$), and at the end time of the creep ($t$-year), and $k$ is a constant grain
growth factor. The grain growth rates are plotted on a logarithmic scale against the reciprocal of $T$
(**Figure 7**). For changes in grain size from the related specimens before and after creep see **Table**
**3**. Correspondingly, the $Q_{gbd}$ values calculated were 41.4 kJ mol$^{-1}$, 40.8 kJ mol$^{-1}$, and 40.9 kJ
mol$^{-1}$ for the specimens at 20 m, 40 m, and 60 m, respectively. These $Q_{gbd}$ values are comparable
to the values of 40.6 kJ mol$^{-1}$ obtained in laboratory experiments on polycrystalline ice (Jumawan,
1972), and 42.4 kJ mol$^{-1}$ from 13 polar firn cores (Cuffey and Paterson, 2010) for grain-boundary
self-diffusion of polycrystalline ice. Further, the ratio of $Q_{gbd}/Q_c$ is 0.67, 0.47, and 0.4 for the 20 m,
40 m, and 60 m specimens, respectively. We noted that the ratio of 0.67 for $Q_{gbd}/Q_c$ was
recommended by Hobbs (1974) and Cuffey and Paterson (2010). The $Q_c$ values calculated using
the Arrhenius relation for the 40 m and 60 m specimens are likely greater than the actual values,
and hence are seemingly less reliable. There is little difference between the two-SRmin-derived
$Q_c$ value (102.8 kJ mol$^{-1}$) and the three-SRmin-derived $Q_c$ value (100.7 kJ mol$^{-1}$), implying
that these two avenues for calculating $Q_c$ have equal utility. Moreover, the above $Q_{gbd}$ values
are lower than the 48.6 kJ mol$^{-1}$ that was inferred by the grain growth rate for firn samples with
densities ranging from 320–650 kg m$^{-3}$ from cores drilled at the South Pole, Antarctic (Gow,
1969), which makes a ratio of 0.67 for $Q_{gbd}/Q_c$ an unreliable sole-criterion. In short, it is difficult
to assess the reliability of both $Q_c$ and $Q_{gbd}$, as discussed above due to their scatter and debates
in the current literature. Thus, these $Q_c$ values estimated in this work, ranging from 61.4–102.8
kJ mol$^{-1}$, are reasonable, aligning with the literature range of 44.8–113 kJ mol$^{-1}$ (**Table 4**).


Table 3. Grain area (mm$^2$) measured from optical thin sections for samples at –5ºC, –18ºC, and –
30ºC from depths of 20 m, 40 m, and 60 m before and after creep.

| Depth | 20 m | | 40 m | | 60 m | |
| --- | --- | --- | --- | --- | --- | --- |
| T/ºC | Before | After | Before | After | Before | After |

| −5 | 0.29±0.25 | 0.42±0.28 | 0.53±0.32 | 0.79±0.67 | 0.78±0.67 | 0.97±0.8 |
|---|---|---|---|---|---|---|
| −18 | 0.29±0.25 | 0.34±0.2 | 0.53±0.32 | 0.7±0.42 | 0.78±0.67 | 0.9±0.59 |
| −30 | 0.29±0.25 | 0.31±0.17 | 0.53±0.32 | 0.57±0.34 | 0.78±0.67 | 0.81±0.56 |

A great challenge is the estimation of the $Q_c$ using the SRmin including the −30$^o$C specimens, whose SRmin shows high variability due to the extraordinarily slow strain rate at low temperatures. This difficulty cannot be resolved by extrapolating experimental data (Sinha, 1978; Hooke et al., 1980), e.g. the use of Andrade's law (Glen, 1955). Instead, we turned our focus to studying the relationship between the SRmin and temperature by constraining our data in a wide range of $Q_c$ values reported in existing literature presented in **Table 4**. Clearly, there is a larger scatter of $Q_c$ values for firn than for ice. The increase of $Q_c$ from mono-crystalline and bi-crystalline to polycrystalline ice implies that the greater the reduction in the constraint from grain boundaries, the greater is $Q_c$. Alternatively, firn creep is easier than that of polycrystalline ice due to either the easier sliding of grains in firn along more directions in the more porous and heterogeneous structure (Section 3.3), or the decrease of viscosity associated with inclusions (e.g.

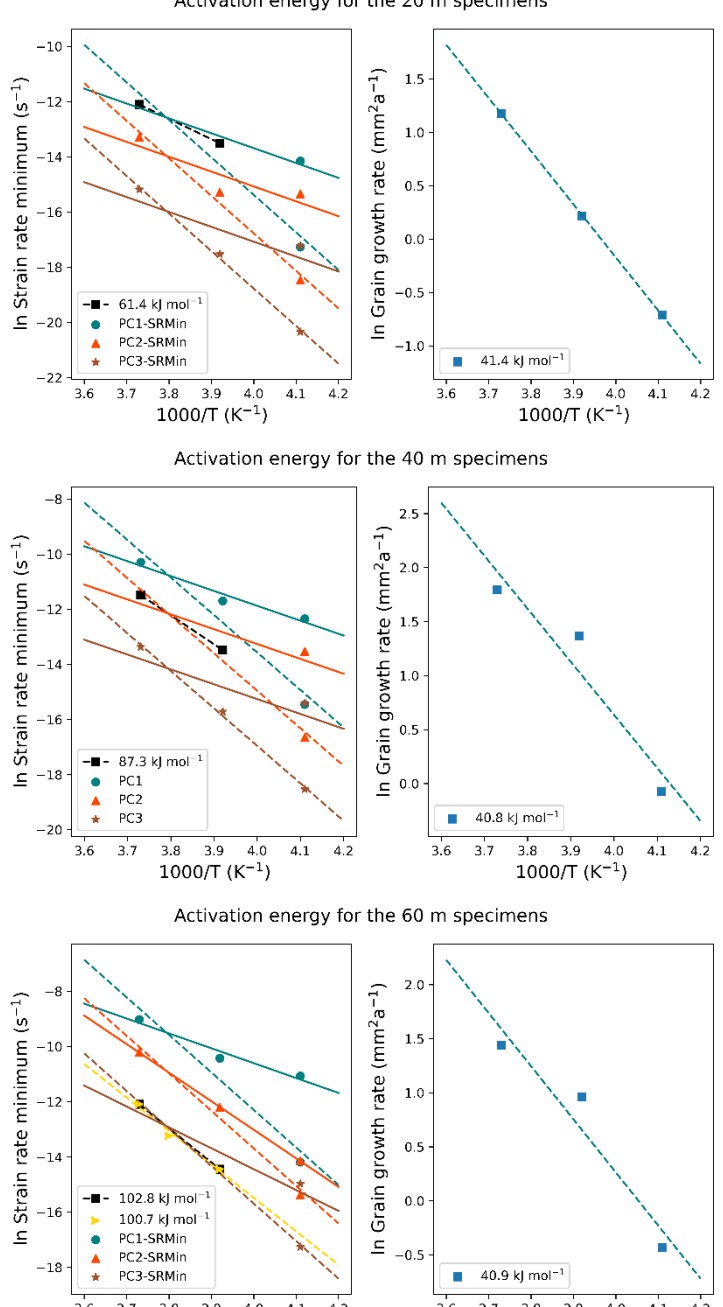

**Figure 7:** Arrhenius plots to estimate the apparent activation energy of creep ($Q_c$; left panel) and

the apparent activation energy of grain-boundary diffusion ($Q_{gbd}$; right panel) from the firn

specimens noted. The teal, orange, and brown solid lines are the upper bound (44.8 kJ mol$^{-1}$) of

PC1-SRmin, PC2-SRmin, and PC3-SRmin, respectively, while the teal, orange, and brown dashed

lines are the lower bound (113 kJ mol$^{-1}$) of PC1-SRmin, PC2-SRmin, and PC3-SRmin,
respectively (Table 2). The teal circles, the orange triangles, and the brown stars are the data in
Table 2. The black dashed lines are from *only* two SRmins at –5°C and –18°C (the black squares
are the data measured), whose $Q_c$ is indicated in each subfigure. The yellow dashed line is from
the three SRmins at –5°C, –18°C in this work, and –10 °C from Li and Baker (2022a) (the yellow
triangles are the measured data), whose $Q_c$ is 110.7 kJ mol$^{-1}$. The blue dashed lines (right panel)
are from grain growth rate at three temperatures (the blue squares are the observed data), whose
$Q_{gbd}$ is indicated in each subfigure.


Table 4. Apparent activation energy for the creep of firn and ice, $Q_c$, reported in literature.

| $Q_c$ kJ mol$^{-1}$ | Sample | Density kg m$^{-3}$ | Temperature °C | Methods | Source |
|---|---|---|---|---|---|
| 58.6–113 | Firn | ~400 | [–13.6, –3.6] | Uniaxial/Hydrostatic Compression | Landauer (1958) |
| 44.8–74.5 | Firn/Bubbly Ice | 440–830 | [–34.5, –0.5] | Uniaxial Unconfined Compression | Mellor and Smith (1966) |
| 54 | Firn/Bubbly Ice | Undetailed | [−28, −16] | Shear Deformation of Boreholes | Paterson (1977) |
| ~72.9 | Firn | 320–650 | Unnecessary | Grain Growth Rate | Gow (1969) |
| 69 ± 5 | Firn | 423 ± 8 | [–19, –11] | Triaxial Compression | Scapozza and Bartelt (2003) |
| ~60 | Artificial/Natural Ice (South Pole) | ~917 | –15 | Torsion Creep Test | Pimienta and Duval (1987) |
| 61 | Polycrystalline Ice | ~917 | –9.6 | Hydrostatic Pressure | Duval et al. (1983) |
| 78 | Monocrystalline Ice | ~917 | [–30, –4] | Derived from Bicrystal Ice | Homer and Glen (1978) |
| 75 | Ice Bicrystal | ~917 | [–30, –4] | Tensile Test Parallel to Grain-boundary | Homer and Glen (1978) |


Baker and Gerberich, 1979; Goodman et al., 1981) that facilitate the intra- and inter-grain sliding
(Salamatin et al., 2009). In principle, $Q_c$ of firn should exceed that for polycrystalline ice.
Intriguingly, some reported $Q_c$ values from firn are less than that for ice, meaning the degree of
spatial freedom in the ice-matrix is limited by the topological structure of the firn (Liu et al.,
2022). Incidentally, the effective stress of porous materials is determined by not only its porosity,
but also other factors, e.g. the microstructural topology (Liu et al., 2022) and the impurity types
and concentrations in the firn. However, this issue is beyond the scope of this work. In seeking a
conclusion, we evaluated the dependence of creep activation energy on firn density. The data
indicate no discernible relationship between these two parameters (**Fig. 8**). In summary, a $Q_c$ for
firn, which ranges from 44.8–113 kJ mol$^{-1}$, is plausible due to the intrinsic nature of natural firn
that has a far more complicated and changeable microstructure than ice.

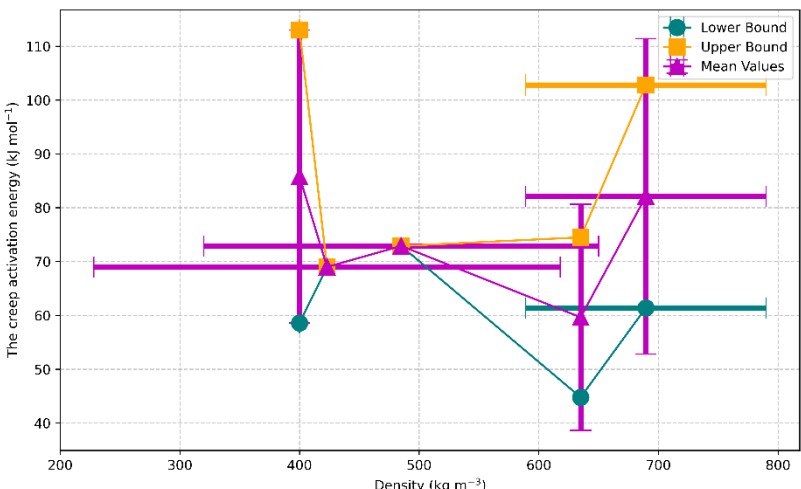


**Figure 8:** Plots of the creep activation energy vs. firn densities. For each density, three values are
shown: the lower bound (minimum activation energy, teal), the upper bound (maximum activation
energy, orange), and the mean value (magenta). Error bars represent the standard deviation of the
mean. Data are sourced from Table 4 and the present study.


The value of the stress exponent $n$ is determined by plotting the line fit the logarithm relation of
the steady-state strain rate, $\dot{\varepsilon}$, versus the effective stress, $\sigma$, and is, thus, the slope of this line
from the measured SRmins (**Table 2**). We determined stress exponent ($n$) values of approximately
0.1 and –1.2 for the –5°C and –18°C samples based on observed data, respectively. This result
directly contradicts the value of $n \approx 4.3$ reported from the same Greenland firn core by Li and
Baker (2022a). Further, these values fall entirely outside the established range of ~1 to ~7.5
(mean ~4.25 ± 3.25) documented across decades of ice mechanics literature (Glen, 1955; Hansen
and Landauer, 1958; Butkovich and Landauer 1960; Kamb, 1961; Paterson and Savage, 1963;
Higashi et al, 1965; Mellor and Testa, 1969; Raymond, 1973; Hooke, 1981; Thomas et al., 1980;
Duval et al., 1983; Weertman, 1983,1985; Azuma and Higashi, 1984; Pimienta and Duval, 1987;
Budd and Jacka, 1989; Jacka and Li, 1994; Goldsby and Kohlstedt, 2001; Bindschadler et al.,
2003; Cuffey, 2006; Chandler et al. 2008; Cuffey and Kavanaugh, 2011; McCarthy et al., 2017;
Millstein et al., 2022; Colgan et al., 2023; Li, 2025). The wide range of reported $n$-values is
governed by a complex interplay of deformation mechanisms—including grain boundary sliding,
diffusion (lattice and grain boundary), and dislocation processes, e.g. hard-slip-dominated,
dislocation-accommodated grain boundary sliding, and grain boundary sliding-limited basal
dislocation—across varying stresses, temperatures, crystalographic fabrics, impurity contents, and
grain-size-to-sample-size ratios. We attribute the significant discrepancy in these findings to the
experimental conditions. The lower temperatures used (down to –30°C) induce slower strain rates,
which prevented the tests from reaching a critical strain rate minimum (SRMin). Therefore, to
accurately estimate the activation energy for deformation, it is necessary first to calibrate the
SRMin value for all noised samples. A constant stress exponent value of $n \approx 4.3$ (Li and Baker,
2022a) was used to compute the activation energy. This necessary simplification—an
acknowledgement of current methodological limitations rather than a dismissal of the underlying
physics—introduces a key uncertainty that highlights the need for future advancements in
observational methodology within firn research. To proceed, the post-calibration SRmins for the –
5°C and –18°C samples are highlighted in **Table 2** (see **Appendix B** in detail). It is important to
note that the stress exponent does not depend on the density of the tested samples, thereby
negating any basis for discussing a relationship between the stress exponent and sample density.
Instead, variations in stress corresponding to density variations are manifested in the strain rate,
ensuring that the derivation of the stress exponent and activation energy remains consistent. From
here on we only discuss the applied stress since there is little difference between the effective
stress and applied stress for calculating the stress exponent (Li and Baker, 2022a). Based on both
the reported range of $Q_c$ and the two observed SRmins at $-5°C$ and $-18°C$, the SRmins for the
$-30°C$ samples are inferred (**Table 2**), using the Arrhenius relation. Also, based on both the
observed and inferred SRmins with the upper and lower bounds (**Table 2**), a series of fitted
functions are then found between the SRmin and the reciprocal of the temperature (°C), $1/T_c$:

20-m samples:

$$
\begin{cases}
\text{SRMin} = -3\times10^{-5}/T_c - 7\times10^{-7}\,[R^2 = 0.988;\,PC1(L\,20)] \\
\text{SRMin} = -3\times10^{-5}/T_c - 2\times10^{-7}\,[R^2 = 1;\,PC1(U\,20)] \\
\text{SRMin} = -1\times10^{-5}/T_c - 3\times10^{-7}\,[R^2 = 1;\,PC2(L\,20)] \\
\text{SRMin} = -9\times10^{-6}/T_c - 2\times10^{-7}\,[R^2 = 0.987;\,PC2(U\,20)] \\
\text{SRMin} = -2\times10^{-6}/T_c - 6\times10^{-8}\,[R^2 = 0.998;\,PC3(L\,20)] \\
\text{SRMin} = -1\times10^{-6}/T_c - 3\times10^{-8}\,[R^2 = 0.976;\,PC3(U\,20)]
\end{cases},
$$

40-m samples:

$$
\begin{cases}
\text{SRMin} = -2\times10^{-4}/T_c - 4\times10^{-6}\,[R^2 = 0.988;\,PC1(L\,40)] \\
\text{SRMin} = -2\times10^{-4}/T_c - 2\times10^{-6}\,[R^2 = 1;\,PC1(U\,40)] \\
\text{SRMin} = -6\times10^{-5}/T_c - 2\times10^{-6}\,[R^2 = 1;\,PC2(L\,40)] \\
\text{SRMin} = -6\times10^{-5}/T_c - 1\times10^{-6}\,[R^2 = 0.987;\,PC2(U\,40)] \\
\text{SRMin} = -1\times10^{-5}/T_c - 3\times10^{-7}\,[R^2 = 0.998;\,PC3(L\,40)] \\
\text{SRMin} = -9\times10^{-6}/T_c - 2\times10^{-7}\,[R^2 = 0.976;\,PC3(U\,40)]
\end{cases},
$$

60-m samples:

$$
\begin{cases}
\text{SRMin} = -7\times10^{-4}/T_c - 2\times10^{-5}\,[R^2 = 0.988;\,PC1(L\,60)] \\
\text{SRMin} = -6\times10^{-4}/T_c - 6\times10^{-6}\,[R^2 = 1;\,PC1(U\,60)] \\
\text{SRMin} = -2\times10^{-4}/T_c - 7\times10^{-6}\,[R^2 = 1;\,PC2(L\,60)] \\
\text{SRMin} = -2\times10^{-4}/T_c - 4\times10^{-6}\,[R^2 = 0.987;\,PC2(U\,60)] \\
\text{SRMin} = -3\times10^{-5}/T_c - 1\times10^{-6}\,[R^2 = 0.998;\,PC3(L\,60)] \\
\text{SRMin} = -3\times10^{-5}/T_c - 7\times10^{-7}\,[R^2 = 0.976;\,PC3(U\,60)]
\end{cases},
$$

where PC1(L20) and PC1(U20) indicate the lower and upper bound values of the post-calibration SRmins from the 20 m samples (**Table 1**), and other symbols are similarly formatted, e.g. PC1(L40), PC1(U40), PC1(L60), PC1(U60), and so on. These relationships are plotted in **Figure 8**, where the SRmin vs. $1/T_c$ plots from the three depths are almost the same shape, implying that the SRmin is dependent on the temperature at a constant stress. It is important to note that the average (minimum) strain rate for the secondary creep stage for a given temperature increases with increasing depth/density of the samples (**Figure 8**; **Table 2**). This is opposite to a decrease of

the SRmin at a fixed stress and temperature in **Figure 8** and **Table 4** in Li and Baker (2022a).
These changes in SRmin are irrespective of the stress (**Appendix A**). The temperature plays a
predominant role during firn creep for a given density of sample at a constant stress. An
interesting question on firn creep at a specific temperature is whether the SRmin slows down or
speeds up with decreasing density of firn. Certainly, natural firn samples raise the complexity in
interpreting the firn creep due to the influences both from inclusions (Li and Baker, 2022a and
references therein; Li, 2024), and from the topology of the microstructures (Liu et al., 2022). In
addition, there is a broad spread of the SRmin at each depth, in which the SRmin varies by several
times, even one order of magnitude or more between the different possibilities of post-calibration
SRmins (**Figure 8**), implying that the microstructure of the sample significantly influences the

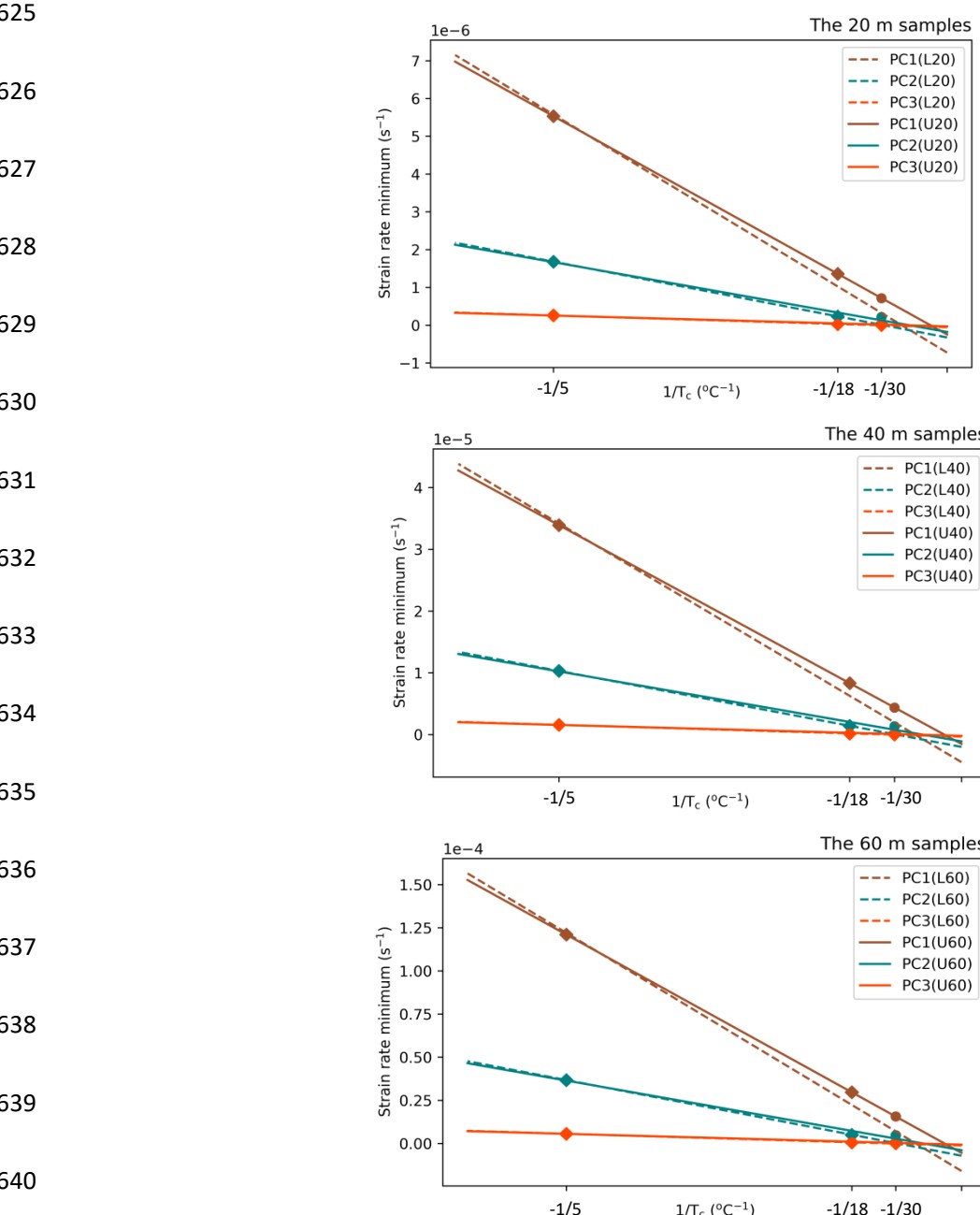

**Figure 9:** Plots of the strain rate minimum versus the reciprocal of temperature. PC1(L20) and PC1(U20) indicates the lower and upper bound, respectively, from the 20 m samples via PC1 as noted in Table 2, and so on. The circles indicate the upper bound data measured and inferred, while the squares indicate the lower bound data. The dashed line is the fit from the lower bound, while the solid line is the fit from the upper bound.

process of the creep of firn. Moreover, it is hard to generalize a universal formula for predicting
the SRmin at temperatures below −30°C, where the SRmins becomes negative (**Figure 8**). Thus,
there is a need for an in-depth understanding of the polar firn creep behavior in secondary creep
stage.

To illustrate the differences between the $Q_c$ values calculated from PC1-SRmin, PC2-SRmin,
and PC3-SRmin, we have plotted them in **Figure 7**. Interestingly, the Arrhenius plots of the
natural logarithm of strain rate with $1000/T$ (**Figure 7**) are similar to those observed by Glen
(1955) and Homer and Glen (1978), implying that there is no significant difference in the creep
mechanism for a temperature range of –30 ºC to –5 ºC (Glen, 1955; Homer and Glen, 1978),
where both diffusion via grain-boundary, vacancy or interstitial defects (Barnes et al., 1971;
Brown and George, 1996; Nasello et al., 2005; Li and Baker, 2022b), and dislocations contribute
to the creep of polar firn.

**4. Conclusions**
Constant-load creep tests were performed on three cylindrical specimens tested from depths of 20
m (applied stress 0.21 MPa), 40 m (0.32 MPa) and 60 m (0.43 MPa) at temperatures of –5 ±
0.2ºC, –18 ± 0.2ºC, and –30 ± 0.2ºC from a firn core extracted at Summit, Greenland in June
2017. The microstructures were characterized before and after creep testing using the micro-CT
and thin sections viewed between optical crossed polarizers. It was found that:

1.  Microstructural parameters measured using the micro-CT show that the polar firn densified

during the creep compression (e.g. from 685 to 729 kg m$^{-3}$ for the 40 m specimen at $-5^{\circ}$C),

*viz.*, the TP (from 25.5 to 20.7%), the ECDa (from 0.86 to 0.69 mm), the SSA (from 3.26 to

3.02 mm$^{-1}$), and the SMI (from $-1.85$ to $-2.8$) decreased, while the S.Th (from 0.95 to 0.99

673        mm) and the CP (from 0.01 to 0.02%) increased. Anomalies in the microstructures, especially

at low temperatures of $-18^{\circ}$C and $-30^{\circ}$C, are likely due to metamorphism under temperature

gradients, the radial dilation effect during firn deformation, the measurement uncertainty of

the micro-CT, or the anisotropy and the heterogeneity of natural firn.

2.  The transient creep behavior of firn at constant stress and different temperatures obeys an

Andrade-like law, but, the time exponent $k$ of 0.34–0.69 is greater than the 0.33 found for ice.

This is due to fewer grain-boundary constraints in porous firn than in ice.

3.  The secondary creep behavior of firn at constant stress and different temperatures presented

here shows that the strain at the SRmin increases with decreasing firn density and increasing

creep temperature. In particular, low-density firn during creep at high temperatures shows that

the strain at the SRmin, e.g. 11.8% and 7.5% respectively from the 20 m and 40 m specimens

at $-5^{\circ}$C, is greater than the strain of 3%, which is the maximum found at the SRMin of ice.

4.  The tertiary creep behavior of firn at constant stress and different temperatures is more easily

observed from lower-density specimens at greater effective stresses and higher creep

temperatures. The strain softening in tertiary creep is primarily due to recrystallization.

5.  The apparent activation energy for the firn creep has a wide range of 61.4–102.8 kJ mol$^{-1}$

because the grains in firn slide more easily along more directions in the more porous and

heterogeneous structure, the enhanced fluidity from inclusions, and the topological structure

of the firn. In addition, the SRmin is a function of the temperature, depending on the microstructure of firn and the inclusion content. The predicted SRmin increases with increasing firn density at a given temperature and is independent of the effective stress. Lastly, there is no significant difference in the creep mechanism at temperatures ranging from $-30^{\circ}$C to $-5^{\circ}$C.

The creep of polar firn behaves differently from full-density ice, implying that firn densification is an indispensable process in fully understanding the transformation of snowfall to ice in the polar regions. Observed firn deformation indicates that temperature plays a determined role in firn densification. Thereby, it will be helpful to bridge a gap between the firn temperature and the climate of the past for reconstructing paleoclimate. Also, it will be helpful to apply a confining load to investigate the microstructure of the creep of polar firn with smaller initial particle sizes at low temperatures using the micro-CT. Further studies of interest are to investigate the quantitative relationship between the microstructural parameters and the mechanical behavior of polar firn, and when the onset of recrystallization occurs during creep, as well as verify the SRmin predicted by the relationship of SRmin vs. temperature from the firn specimens at more extensive ranges of stresses and temperatures.

**Appendix A:** Hydrostatic pressure, the applied stress, and the effective stress

The hydrostatic pressure, p, which varies with temperature, along with the cohesion of the ice and the friction angle between snow particles, plays a significant role in determining the apparent activation energy and, consequently, the strength of the ice (Fish, 1991). It was calculated from the overburden pressure of snow, i.e. $p = \bar{\rho}_f gh$, where $\bar{\rho}_f$ is the average firn density above the depth of interest, *h*, and *g* is the acceleration of gravity. At Summit, p at the depths of 20 m, 40 m, and 60 m was estimated to be ~0.1 MPa, ~0.22 MPa, and ~0.38 MPa, respectively. Note that the slope of the surface of ice sheets and glaciers at Summit is idealized to be zero, i.e., their surfaces are horizontal. The applied stress, $\sigma$, is the applied load divided by the cross-sectional area of a sample. The $\sigma$ at the depths of 20 m, 40 m, and 60 m were 0.21 MPa, 0.32 MPa, and 0.43 MPa, respectively. The effective stress, $\tilde{\sigma}$, is defined as $\sigma$ divided by the fraction of ice matrix in firn, see in detail from Li and Baker (2022a). Thereby, $\tilde{\sigma}$ is 0.32 MPa (the mean porosity of 34.9%), 0.43 MPa (24.8%), and 0.5 MPa (14.4%) from the 20–60 m samples, respectively. Note that the stresses were vertically loaded on the sample (parallel to the direction of core axis of the sample) in laboratory tests. Ideally, in order to be analogous to the densification of firn in nature, $\tilde{\sigma}$ for laboratory samples from a given depth should be equal to the p of firn *in situ* at an equivalently same depth at Summit, namely $\tilde{\sigma}$/p = 1. However, in consideration of the laboratory timeframe for experiments (Pimienta and Duval, 1987), the stresses applied in laboratory tests are usually higher with a resulting higher rate of deformation than those *in situ*. Thus, to observe the effect of the stress on the creep of firn with different densities at different depths, we designed the following configuration of the $\tilde{\sigma}$/p with depth, *viz*., 0.32 MPa/~0.1 MPa = ~3.2, 0.43 MPa/~0.22 MPa = ~1.95, and 0.5 MPa/~0.38 MPa = ~1.32 for the samples from the depths of 20 m, 40 m,

and 60 m, respectively. In this manner, the decrement of $\tilde{\sigma}$/p with increasing depth represents
the decrease of the effective stress with increasing depth. Also, it's important to note that the
strain rates achieved during creep experiments in laboratory settings are 6 to 7 times faster than
on ice sheets due to the constraints of conducting experiment in reasonable times, which requires
higher loads.

**Appendix B:** Strain rate minimum inferred via two kinds of constraints
To improve the reliability of inferred SRmins, two kinds of constraints were applied. First, the
SRmins from the –5ºC and –18ºC samples are calibrated using Glen's law $\dot{\varepsilon}=A\sigma^n$ with $n = 4.3$
(Li and Baker, 2022a). PC1-SRmin, PC2-SRmin, and PC3-SRmin indicate three possibilities of
the SRmins that are calculated from the 20 m, 40 m, and 60 m samples via the *only* SRmin
observed at a given temperature (Table 2). As an example, for the –5ºC samples, there exist three
possibilities from three depths. 1) The SRmin observed from the 20 m sample in bold italic font is
used to calculate two other SRmins for the 40 m and 60 m samples in the italic font in the column
of PC1-SRmin. 2) In the same manner as in scenario 1), the SRmin observed from the 40 m
sample is calculated in the column of PC2-SRmin in the bold italic font, and the SRmin observed
from the 60 m sample is calculated in the column of PC3-SRmin in the bold italic font. 3) In the
same manner as in scenarios 1) and 2), the SRmin is calculated for the –18 ºC samples in turn
from three depths. Second, the SRMin of the –30ºC samples is inferred on the basis of the range
of $Q_c$, i.e., from 44.8 kJ mol$^{-1}$ (upper bound)   to 113 kJ mol$^{-1}$ (lower bound), using the Arrhenius
relation.

**Data availability**

The data supporting the conclusions in this study are available at https://arcticdata.io/catalog.

**Author contribution**

Y.L. and I.B. designed the experiments and Y.L. carried them out. Y.L. analyzed the data and visualized the relevant results. Y.L. prepared the manuscript with contributions from all co-authors (K.K. and I.B.).

**Competing interests**

At least one of the (co-)authors is a member of the editorial board of The Cryosphere.

**Acknowledgements**

This work was sponsored by the National Science Foundation under Arctic Natural Science Grant No. 1743106. Y.L. gratefully acknowledges Ciao Fu for her great support and help in during the COVID-19 pandemic. The authors wish to thank Chris Polashenski, Zoe Courville and Lauren B. Farnsworth at USA-CRREL for their assistance with the storage of the firn cores. We also acknowledge the use of facilities of the Ice Research Laboratory (Director-Erland Schulson) at Dartmouth College. Finally, the authors thank Editor Nanna Bjørnholt Karlsson, Louis Védrine and an anonymous reviewer for their constructive comments, which significantly improved the quality of this manuscript.

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
