# Peer review of "Observations of creep of polar firn at different temperatures"

_EGUsphere, 2024_

## Author Comment (AC1)

Dear Dr. Védrine,

We thank you very much for your comments to improve the manuscript.
Please find your comments below in blue font and our responses in black font.

The authors introduce a method to quantify the influence of temperature on firn creep. Through laboratory creep tests conducted at various temperatures and with different initial microstructures, they investigate how firn responds under these conditions. The microstructure of the samples is assessed before and after the tests using microtomography and thin-section analysis. Subsequently, the activation energy is determined and compared with the activation energy for grain-boundary sliding, which is estimated based on the observed grain growth rates.

This study represents a notable advancement in modelling the mechanical behaviour of firn, enhancing our understanding of ice material properties and informing the interpretation of ice-core data relevant to paleoclimatology.

However, the methodology used to determine the activation energies is not sufficiently detailed (lines 385–390). Specifically, the authors assume a fixed value for the stress exponent (without providing the actual value) and neglect to mention that the pre-factor A is considered constant across different microstructures and test temperatures. These omissions represent significant methodological shortcomings. I have serious concerns about the validity and applicability of the methodology, raising doubts about the reliability of the results. Therefore, I recommend major revisions.

Thank you for these comments, which we've used to improve the clarity of our manuscript. Please find our detailed explanations after each of your comments below.

General comments:

To determine the activation energy, the authors use the Glen-type power law (line 386). This equation introduces the activation energy (Qc), the stress exponent (n), and the pre-factor (A). Thus, to identify the value of Qc, assumptions about A and n must be made

Prefactor: The authors assume that the pre-factor in the power law remains constant across the different temperatures tested. However, the sample densities vary from 589 kg/m³ to 615 kg/m³ at 20 m depth. This approximation is not mentioned by the authors and must be acknowledged as a limitation of the method.

The pre-factor (*A*) in the Glen's flow law is typically regarded as a constant in studies of firn and ice mechanics to derive stress exponents and activation energy. Notably, this assumption has often been omitted in the literature, e.g. Goldsby & Kohlstedt

(1997; 2001). For completeness, we've included the following sentence in Section 3.5 to describe the role of $A$ used in our calculations:

*"Based on Glen-King's results in deriving the activation energy (Glen, 1955)*

$\dot{\varepsilon} \quad A \quad Q_c/RT) = B\sigma^n \exp(-Q_c/RT)$, *the pre-factor A, the material parameter B*

*(Glen, 1955; Goldsby & Kohlstedt, 2001), and the stress exponent n (Li and Baker, 2022a) are assumed to be constant as performed in the literature."*

In this research, samples with densities ranging from approximately 550 to 830 kg/m³ are utilized to investigate the deformation of firn plasticity, aligning with a widely accepted power law deformation mechanism (Li & Baker, 2022a). A significant challenge in experimental science arises from discrepancies between theoretical predictions and laboratory results, particularly when using natural samples from Greenland and Antarctica. However, the observed density variation between 589 and 615 kg/m³ at a depth of 20 m is deemed acceptable for mechanical experiments involving natural porous samples. These variations stem from multiple factors, including the intrinsic properties of the samples, e.g. inclusions (impurities, dust, bubbles, clathrate hydrates), the effects of deformation and partial annealing of firn due to stress distribution and temperature changes during drilling, extraction, transportation, or storage, and also the fact that the samples are taken from adjacent parts of the core, and might sample heterogeneous density layers, as well as potential measurement errors associated with the equipment used. To clarify this point, we've add the following sentence to Section 3.5:

*"The variability in density for the samples from a depth of 20 m on the mechanical behavior are negligible due to a small difference (up to ~4%), between them, which falls within an acceptable error range in the literature. This is likely related to multiple factors, including the intrinsic properties of the samples, e.g. inclusions (impurities, dust, bubbles, clathrate hydrates), the effects of deformation and partial annealing of firn due to stress distribution and temperature changes during drilling, extraction, transportation, or storage, and the fact that the samples are taken from adjacent parts of the core, and might sample heterogeneous density layers, as well as potential measurement errors associated with the equipment used."*

Stress exponent: The stress exponent is only mentioned toward the end of the section (lines 462-470), where the values 0.1 and -1.2 are considered. However, the method for determining these values is not explained. Moreover, these exponents are inconsistent with those reported in the literature and in Li and Baker (2022), and they do not align with any known physical behaviour of materials.

We agree with you that the stress exponent values of n = 0.1 and n = -1.2 are inconsistent with the literature, and we tried to highlight that point in Lines 463–464:

*This is in disagreement with the reported n = ~4.3 by Li and Baker (2022a)*. To make this point clearer, we modified the text to:

*"We found n to be ~0.1 and ~-1.2 for the -5°C and -18°C samples, respectively, which is in disagreement with the reported n = ~4.3 by Li and Baker (2022a). This significant discrepancy implies that the uncalibrated SRMin value from all the samples is not appropriate for estimating the stress exponent, and hence the activation energy during their deformation."*

To clarify our methods in this section, we added the following text to describe our method of obtaining the stress exponent:

*"The value of n is determined by plotting the line fitted the logarithm relation of the steady-state strain rate, $\dot{\varepsilon}$, versus the effective stress, $\sigma$, thereby being the slope of this line."*

A "post-calibration" method is then introduced, which imposes a fixed stress exponent but fails to account for density dependence. This approach leads to variable results, depending on the chosen reference sample. These inconsistencies arise from the identification of the power law using data in which both stress and density vary simultaneously. As demonstrated in Li and Baker (2022a), the strain-rate minimum (SRMin) is dependent on density, with the strain rate decreasing by a factor of 12 when the density increases from 756 to 861 kg/m³. However, the effect of microstructure is overlooked in this study, as it treats samples with densities ranging from 589 to 790 kg/m³ as identical.

Stress exponents reported during the creep of polar firn range from 4.1 ± 0.37 to 4.6 ± 0.16 (Li and Baker, 2022a). It is crucial to emphasize that the stress exponent does not depend on the density of the tested samples, thereby negating any basis for discussing a relationship between the stress exponent and sample density. Instead, variations in stress corresponding to density will manifest in the strain rate, ensuring that the derivation of the stress exponent and activation energy remains consistent, as noted by the reviewer. Further, the minimum strain rate is indeed influenced by density (Li and Baker, 2022a), which is typically utilized to derive the stress exponent in accordance with Glen's law, considering the effective stress's impact on porous firn reflected in strain rates. Consequently, the stress exponents are expected to be similar across samples of varying densities. To highlight this distinction in the manuscript, we added the following text:

*"It is important to note that the stress exponent does not depend on the density of the tested samples, thereby negating any basis for discussing a relationship between the stress exponent and sample density. Instead, variations in stress corresponding to density manifest in the strain rate, ensuring that the derivation of the stress exponent and activation energy remains consistent."*

The authors need to improve the methodology and clearly outline the assumptions made, particularly regarding the density dependence of viscoplastic behaviour. This could be based on their previous work (Li and Baker, 2022) or by considering other models from the literature. Finally, the discussion in the "activation energy" section should be revisited in light of these methodological assumptions.

Thank you for pointing out our omissions in the methodological steps described above. In addition to the relevant methodological assumptions we've added above, we also included the following description of the method to calculate Qc:

*"The value of $Q_c$ is equal to the slope of a line fitted $\ln \dot{\varepsilon}$ versus 1/T as did in Goldsby & Kohlstedt (1997; 2001)."*

Specific Comments:

Lines 43-48: What about the study of Burr et al., (2019) for the in-situ compression test? Does it relate or include relevant data to evaluate the results of this study?

We chose to not include this reference in our introduction. Burr et al. (2019) extensively examined the influence of pore closure and ice crystal grain growth on the densification of polar firn, utilizing *in situ* micro-computed tomography imaging while neglecting airflow effects. Although their study addressed thermal treatments of samples 801, 806, and 901, the effects were predominantly attributed to pressureless sintering, as these samples underwent strain without an applied load. Sample 806 was initially compressed at –10°C before being thermally treated at –2°C. In contrast, our research focuses on the deformation of firn samples subjected to constant stress across varying temperatures.

Lines 57-59: Using homogenization approaches, considering the behaviour of ice, is common for studying the physical properties of firn and snow.

Yes, porous snow and firn primarily consist of an ice matrix interspersed with air. Hence, research on their mechanical properties often centers on the solid ice component, while also considering the influence of airflow.

Lines 233-235: Please provide more details on how the thermal gradient was evaluated during your experiments.

We've added the following text to address how the thermal gradient was generated:

*"The thermal gradient is likely related to the inherent fluctuation of 0.5$^o$C around a test temperature due to the thermometers' accuracy, thereby thermal sensitivity-heightened temperature cycling within the firn (Mellor and Testa, 1969; Weertman, 1985)."*

Lines 285-289: As deformation mechanisms are not directly measured in this study, please add references to the literature in this discussion.

We added the following relevant references to the text:

*"The transient creep stage may be caused by strain hardening that occurs from the yield point to the ultimate strength (Glen, 1955; Jacka, 1984). The plastic deformation is accommodated by an increase in dislocation density through dislocation multiplication or the formation of new dislocations (Frost and Ashby, 1982; Duval and others, 1983; Ashby and Duval, 1985), which leads to an increase of the firn strength as the dislocations become pinned or tangled, and thus more difficult to move."*

Line 336: Unclear, it is the temperature which is a state variable of the strain-rate.

In these experiments, strain rate is considered a state variable of temperature. Under optimal conditions, the strain rate of a creep sample subjected to a specific stress to achieve a certain strain can be exclusively determined by the temperature.

Lines 410-411: The word "methods" can be misleading. Using 2 or 3 data points to identify a parameter is not a separate method. Either remove the word "method" or include the data point at -10 °C in the overall dataset.

We replaced "methods" with "avenues".

Lines 453-455: The statement about the activation energy of firn should be nuanced. While older studies show lower values than those of ice, you have already discussed that values of Qc are highly scattered and debated (as mentioned in lines 416-426 of your manuscript).

Indeed, the activation energies derived in this work exhibit a wide range, which are consistent with a broad spread of 58.6–113 kJ/mol estimated by Landauer (1958). We've highlighted the factors that likely contribute to differences in $Q_c$ values from the older literature in Table 3, e.g. the sample density, temperature inaccuracies during testing, and differences in the methodologies used for derivation. For further clarification, we explained this in the text:

*"The increase of $Q_c$ from mono-crystalline and bi-crystalline to polycrystalline ice implies that the greater the reduction in the constraint from grain-boundaries, the greater is $Q_c$. Alternatively, firn creep is easier than that of polycrystalline ice due to either the easier sliding of grains in firn along more directions in the more porous and heterogeneous structure (Sect. 3.3), or the decrease of viscosity associated with inclusions (e.g. Baker and Gerberich, 1979; Goodman et al., 1981) that facilitate the intra- and inter-grain sliding (Salamatin et al., 2009)."*

Lines 475-480: It's not clear that each brace corresponds to a depth. Please clarify it.

To further indicate what each bracket refers to, we added "20-m samples; 40-m samples; 60-m samples" above their respective bracket.

Technicals Comments:

Figure 6: Please specify in the title of the y-axis in Figure 6 that this is the logarithm of SRmin, to ensure consistency in the names used.

Corrected.

Figure 6: The text and colours in the caption do not appear to correspond to the figure. Please check.

Corrected.

Sincerely,
Yuan Li, Kaitlin Keegan, Ian Baker

---

## Author Comment (AC2)

Dear Reviewer,

We thank you very much for your comments to improve the manuscript.
Please find our responses below in black font, and your comments in blue font.

General comments

This manuscript investigates the metamorphism and deformation mechanism using natural firn samples recovered at Greenland summit by mechanical tests and microstructure observations. Based on the experimental results, activation energy for creep deformation and grain boundary diffusion is estimated. The authors compare the results with previous studies on activation energy and discuss the firn deformation, and differences between firn and solid ice, and argued that the minimum strain rate is determined by temperature. Microstructures of firn samples before and after creep experiments are analyzed by X-ray micro computed tomography. Changes in geometric structure during creep deformation are investigated in detail.

This manuscript provides interesting results in mechanical behavior of firn samples (strain rate vs strain) and extensive 3D data on geometric structures before and after creep experiments. They are important data for discussion the deformation mechanisms and microstructural evolution of firn.

However, I have some significant concerns in the methodology, interpretation and references. In particular, experimental samples and conditions should be verified. Cited references are biased toward the author's paper. Please cite the references widely. Reconstruction of the manuscript is required. Therefore, I recommend major revisions.

Thank you for your thoughtful comments, which greatly improved our manuscript. Please find our detailed responses to the general comments you raised above in our responses to your specific comments below. As noted below in response to multiple comments, we added other references throughout the manuscript wherever possible.

Specific comments:

Abstract and Introduction:

It is difficult to understand the new findings of this study. In the field of ice and snow deformation, it is widely recognized that temperature is an important factor, and that tertiary creep is driven by recrystallization (Cuffey and Paterson, 2010; Faria et al., 2014). Compression deformation of firn, accompanied by an increase in density, differs from that of ice. Different creep behaviors between firn and ice could be

expected.

This study presents novel firn creep data for three different depths of the Summit 2017 core at three different temperatures. As firn deformation studies are scarce, these data provide important empirical data that are useful for improving firn flow laws and models as well as our understanding of the mechanisms driving firn creep. While we expect firn and ice to display different creep behavior, it is still informative to compare them, especially because firn contains an ice-matrix. To emphasize the novelty of this data set, we added the following text to the abstract:

*"The results of these experiments comprise a novel data set of firn creep at three depths of a firn column under three different temperatures, providing useful calibration data for firn model development."*

The Introduction Section includes few references to previous research on firn deformation and metamorphism, making it unclear how this study fits within the context of current research and its problems. In addition to prior studies on ice deformation experiments, please also cite prior studies on firn deformation experiments.

We added a more thorough description of prior deformation studies with the following text:

*"Numerous studies of firn and ice deformation have been conducted (e.g. Steinemann, 1954; Glen, 1955; Landauer, 1958; Mellor, 1975; Salm, 1982; Maeno and Ebinuma, 1983; Jacka, 1984; Ambach and Eisner, 1985; Budd and Jacka, 1989; Li et al., 1996; Meussen et al., 1999; Petrenko and Whitworth, 1999; Bartelt and von Moos, 2000; Jacka and Li, 2000; Durham et al., 2001; Goldsby and Kohlstedt, 2001; Hooke, 2005; Song et al., 2006a, 2006b, 2008; Theile et al., 2011; Treverrow et al., 2012; Hammonds and Baker, 2016, 2018; Li and Baker, 2021, 2022a), but there are few reports about the mechanical behavior at different temperatures. Temperature is a key component of firn and ice-flow models, as the deformation of firn, polythermal glaciers, and temperate glaciers is significantly influenced by the temperature."*

In the Discussion section, there are many comparisons with the authors' own related papers, and the discussion with other studies is not sufficient. Please specify how the findings of this study advance our current understanding of firn deformation.

We've added additional discussions from various sources to Section 3.4:

In Section 3.3, significant references are cited from Li and Baker (2022a), including works by Glen (1955), Landauer (1958), Glen and Jones (1967), Jones and Glen

(1968), Barnes et al. (1971), Homer and Glen (1978), Meussen et al. (1999), Freitag et al. (2002), and Theile et al. (2011). To avoid unnecessary repetition, these references are not elaborated upon in this study.

"*These values of strain at different SRMin values are different from those usually observed at strains of 0.5–3% for fully-dense ice (Cuffey and Paterson, 2010 and references therein), implying that different mechanical behavior between firn and pure ice (Duval, 1981; Mellor and Cole, 1983; Jacka, 1984; Li et al., 1996; Jacka and Li, 2000; Song et al., 2005, 2008; Cuffey and Paterson, 2010)....Additionally, tertiary creep occurs both during quasi-steady state deformation (from the –5$^o$C specimens at 40 m and 60 m) and in the ascending stage (from the –5$^o$C and –18$^o$C specimens at 20 m and the –18$^o$C specimen at 40 m) more easily with lower firn density, greater effective stress, and higher creep temperature, e.g. from the –5$^o$C specimens at 20 m, where the strain softening is primarily due to either recrystallization (Duval, 1981; Jacka, 1984; Jacka and Li, 2000; Song et al., 2005; Faria et al., 2014) or the activated easy slip systems (Jonas and Muller, 1969; Duval and Montagnat, 2002; Alley et al., 2005; Horhold et al., 2012; Fujita et al., 2014; Eichler et al., 2017).*"

The work presented in this manuscript builds upon Li & Baker (2022a) by investigating the impact of temperature on firn creep by conducting deformation experiments on samples from three depths of the Summit 2017 core at three different temperatures. Prior work only investigated the creep of the Summit 2017 firn at different depths, holding all other experimental variables constant. Thus, we believe the thermal focus of this work differentiates it from the prior study.

2. Sample and measurements

Please provide a schematic diagram of experimental setup even if it is shown in supporting paper (Li and Baker, 2022a).

As suggested, we now include Figure 1 from Li and Baker (2022a) to show the experimental setup.

Differences in initial conditions of each sample may significantly impact the results. For example, factors like fabric and impurity concentration may vary with depth. In the case of EastGRIP, it has been reported that fabric develops even in near-surface snow (Montagnat et al., 2022). Although geometric structure is discussed in the text and Table 1, it is also necessary to examine other elements of the initial samples, such as fabric, impurity concentration, and grain size. Not only is there a difference in the initial microstructure depending on the depth, but there is also a heterogeneity unique to the natural sample at the same depth. Otherwise, direct comparisons between

different samples may not be valid. I have question about the reproducibility of the experiment, in particular, strain rate vs strain.

Indeed, any underlying differences in fabric, grain size, and impurity content in firn samples may significantly impact results, highlighting a significant challenge in conducting creep experiments on natural firn. To limit the possibility of significant differences in those variables, care was taken to extract the three replicate samples from each depth as closely as possible to each other. With the limited amount of ice available at each depth in any given core, it is challenging to generate more creep experiment samples and more strain rate vs. strain data. To highlight these points, we've added the following text to Section 2:

*"It's important to note that firn is a heterogeneous material that can have variations in layering, fabric, grain size, and impurity concentration across short distances. Thus, care was taken to extract the three replicate samples from the core at each depth as closely as possible to reduce the variability in their initial conditions."*

3.4 Relationship of strain rate to strain and 3.5 Apparent activation energy for creep:

I have questions about the calibration of experimental data. If the number of experiments is increased or experimental conditions are changed, then no calibration would be necessary. Or is it common practice to make calibration in firn deformation experiments? Looking at Table 2 and Figure 6, 7, it appears that the results vary greatly depending on the type of calibration. The discussion of strain rate (creep curve) and activation energy does not seem robust because of the large influence of the calibration. Please explain clarify, as it is difficult to understand the necessity and appropriateness of the calibration.

The increasing number of experiments cannot guarantee that the results used for deriving the apparent activation energy for creep are not calibrated, as performed by Glen (1955) for developing the Glen flow law. The necessity of calibration in this study will be detailed for your following concern about this sample question.

The variability in the activation energy for firn creep in Figure 6 is consistent with those reported in the literature, thereby ensuring the projected relationship between the strain rate minimum and temperature in Figure 7. Thus, this calibration is necessary, and the calibration method used is appropriate.

Appendix A:

The authors determine the loading stress during deformation experiments from the hydrostatic pressure at the point where the sample was taken, but is this reasonable?

Hydrostatic pressure is considered to have no effect on strain rate in ice (Rigsby, 1958; Cuffey and Paterson, 2010), and in ice it is the deviatoric stress that determines strain rate.

Hydrostatic pressures resulting from the overburden of overlying strata are frequently estimated through the integration of depth-density profiles. This pressure is crucial for determining the depth at which the stress applied to a single sample in laboratory tests corresponds to the conditions within the firn or ice core. The deviatoric normal stress is roughly equal to the difference between the principal stress at a given depth along the ice sheet's surface-base and its hydrostatic pressure, influencing the strain and strain rate during the deformation of snow, firn, and glacier ice. In contrast, the deviatoric shear stress is the same as the non-deviatoric component (Hook, 2005). Consequently, in many regions of polar ice sheets, a low deviatoric stress, typically around 0.1 MPa or less, prevails (e.g. Montagnat et al., 2015). However, the samples used in this study originate from the horizontal surface of the Greenland Summit, where the deviatoric stress approaches zero, indicating that the principal stress is nearly equal to the hydrostatic pressure at the corresponding depth.

It is understandable that a high stress is necessary to make the experiment (deformation) proceed quickly and that the ratio of effective stress to hydrostatic pressure should be considered to approximate actual ice sheet conditions (set so that the effective stress becomes smaller as the depth increases). However, the strain rates obtained in this experiment are on the order of $10^{-5}$ to $10^{-6}$ s$^{-1}$, which is several orders of magnitude larger than actual ice sheet firn. As an example, Faria et al. (2014) estimated the vertical strain rate of EDML firn at 50 m depth as order of $10^{-11}$ - $10^{-12}$ s$^{-1}$. Furthermore, they concluded that EDML firn at 50-m depth is determing in the tertiary creep with dynamic recrystallization.

The high stresses (or high strain rate) will also cause dislocation accumulation and tangle, and recrystallization to be more active than it actually is.

Yes, you are correct. You are highlighting another challenge in conducting laboratory creep experiments on natural firn (and ice). Unfortunately, natural deformation in ice sheets occurs at rates that are orders of magnitude slower than what is possible to achieve in the laboratory setting. These issues are present in all laboratory-based creep studies and must be considered when discussing their results. Thus, we put the following text in Appendix A to remind readers of this caveat:

*"Also, it's important to note that the strain rates achieved during creep experiments in laboratory settings are 6 to 7 orders of magnitude faster than on sheets due to the constraints of running a reasonable experiment."*

Others

L233-235: Could a decrease in density associated with deformational compression occur in a real ice sheet firn?

No. To highlight this point, we've added the following description in the text:
*"In the relatively simple deformation system found at ice-sheet dome sites, such as Summit, there is no mechanism to decrease density during deformational compression. At sites closer to the ice sheet margins, cracking due to extension of the ice may cause a localized decrease in density due to deformation."*

L239-241: Does the fact that the ratio of effective stress to hydrostatic pressure in the experiments (discussed in Appendix A) varies from sample to sample (depth to depth) not affect the differences in density increase?

No. Both the effective stress and the hydrostatic pressure take density, and therefore the porosity/ice-matrix, into account. Thus, the values of effective stress and hydrostatic pressure are proportional to the sample densities at each depth.

L250-255: Only one example of grain size change before and after creep experiment is shown (40-m sample at -5 $^{o}$C). In the manuscript, it just says, refer to Li and Fu (2024) (L401) for other samples, but it needs to mention in the present paper. Please provide other measurement results in grain size changes before and after deformation.

The 40-m sample at -5 $^{o}$C is to show the occurrence of recrystallization during firn deformation, not for all samples. Thus, it is taken as an example. Additionally, we've added the relevant grain size data in Table 3.

*Table 3. Grain area (mm$^2$) measured from optical thin sections for samples at $-5^{o}$C, $-18^{o}$C, and $-30^{o}$C from depths of 20 m, 40 m, and 60 m before and after creep.*

| Depth | 20 m | | 40 m | | 60 m | |
|---|---|---|---|---|---|---|
| $T/^{o}C$ | Before | After | Before | After | Before | After |
| $-5$ | $0.29\pm0.25$ | $0.42\pm0.28$ | $0.53\pm0.32$ | $0.79\pm0.67$ | $0.78\pm0.67$ | $0.97\pm0.8$ |
| $-18$ | $0.29\pm0.25$ | $0.34\pm0.2$ | $0.53\pm0.32$ | $0.7\pm0.42$ | $0.78\pm0.67$ | $0.9\pm0.59$ |
| $-30$ | $0.29\pm0.25$ | $0.31\pm0.17$ | $0.53\pm0.32$ | $0.57\pm0.34$ | $0.78\pm0.67$ | $0.81\pm0.56$ |

L285-294: The strain rate transition (creep curve) in deformation and recrystallization have been described by numerous papers and textbooks (e.g., Budd and Jacka, 1989; Cuffey and Paterson, 2010; Faria et al. 2014). Please cite references widely as well as the authors' papers.

We modified as below:

*"The transient creep stage may be caused by strain hardening that occurs from the yield point to the ultimate strength (Glen, 1955; Jacka, 1984). The plastic deformation is accommodated by an increase in dislocation density through dislocation multiplication or the formation of new dislocations (Frost and Ashby, 1982; Duval and others, 1983; Ashby and Duval, 1985), which leads to an increase of the firn strength as the dislocations become pinned or tangled, and thus more difficult to move. The initial decrease of creep rate may also be related to the rearrangement of dislocations into a more stable pattern through a dragging mechanism (Weertman, 1983) for the $-5^{o}C$ specimens. The tertiary creep stage may be associated with strain softening deriving either from the thermally-activated processes at the high homologous temperature approaching the melting point of ice, or from recrystallization (Li and Baker, 2022a)".*

L306-309: What is the reason why the 20m and 60m samples with large density differences are close to each other and the 40m sample is greater than that?

To make further clarification, we've modified the following description in the text:

*"Interestingly, an evident relationship between the density of firn and the k values, regardless of the effect of stress (Li and Baker, 2022a) and temperature, remains unknown."*

L309-310: I did not understand this logic (These k values imply that the more the constraints from the grain-boundaries, the slower the deformation rate will be,..). Please explain in detail.

To make further clarification, we've modified the following description in the text:

*"A greater k value signifies swifter deformation. These k values derived for firn are generally higher than those for polycrystalline ice, implying that higher firn deformation rate than ice is related likely to its less grain-boundary constraints with more free void space (Li and Baker, 2022a; Li, 2023b)."*

L327-329: I did not understand this logic (likely suggesting that the effect of temperature overwhelmed the effect of impurities during creep of polar firn.). Please explain in detail.

To make further clarification, we've modified the following description in the text:

*"k values lower than 0.33 observed under constant load and temperature occurred at the relatively low effective stresses (Li and Baker, 2022a). In contrast, this is seemingly to occur at the relatively low temperatures due to the steady decrease of k values from –5$^o$C to –18$^o$C, thereby remaining further investigation."*

L369-370: Why does the strain at which the minimum strain rate is achieved vary with density and temperature?

We tried to highlight a reason behind this phenomenon in Lines 371–372:
*"Overall, the strain at the SRMin is greater with lower density and higher temperature, e.g. 11.8% strain from the –5$^o$C specimens at 20 m, 4.1% strain from the –18$^o$C specimens at 40 m, where a larger strain was caused by the longer-lasting strain hardening (Li, 2023b)."*

To make this point clearer, we modified the text to:
*"Overall, the strain at the SRMin is greater with lower density and higher temperature, e.g. 11.8% strain from the –5$^o$C specimens at 20 m, 4.1% strain from the –18$^o$C specimens at 40 m. This is related to the effect of strain hardening on density and temperature (Li, 2023b)."*

L462-464: Why does the stress exponent obtained in this experiment differ from previous studies? The value of 0.1 obtained in this study seems quite low. The difference may be too large to address with calibration alone. Please also cite previous studies other than Li and Baker (2022a) that estimated the stress exponent.

This is due to the inappropriate use of the strain rate minimum for the –30$^o$C specimens, which is difficult to observe a steady-state secondary creep at such low temperature, thereby leading to the different values of the stress exponent from those in Li and Baker (2022a), which exhibits similar flow law from a same core.

We tried to highlight that point in Lines 463–464:

*This is in disagreement with the reported n = ~4.3 by Li and Baker (2022a).* To make this point clearer, with additional reports regarding the stress exponent, we modified the text to:

*"We found n to be ~0.1 and ~-1.2 for the -5$^o$C and -18$^o$C samples, respectively, which contradicts the reported n = ~4.3 by Li and Baker (2022a) and other values around 3 (Glen, 1955; Kamb, 1961; Raymond, 1973; Thomas et al., 1980; Weertman, 1985; Goldsby and Kohlstedt, 2001; Cuffey, 2006). This significant discrepancy implies that the uncalibrated SRMin value from all the samples is not appropriate for estimating the stress exponent, and hence the activation energy during their deformation."*

Also, is it possible for the stress exponent to be negative? There may be large fluctuations in the strain rate obtained in the deformation experiments, which could hinder accurate estimation. If these values are correct, what deformation mechanism do they correspond to?

No. See response above.

I question the practice of determining the stress exponent from experimental results of different samples. If the initial conditions of the samples differ, the deformation characteristics will also change, making it impossible to accurately determine the stress exponent.

Natural firn and ice samples provide a more accurate representation of the ice flow law governing glaciers and ice sheets. According to Glen's flow law, numerous laboratory experiments have consistently yielded a stress exponent value around 3, based on tests utilizing laboratory-generated snow and ice samples with varying initial grain sizes, crystallization preferred orientations, densities, or impurity levels (Cuffey and Paterson, 2010 and references therein). Therefore, selecting initial samples with diverse microstructural parameters is essential for a deeper understanding of flow rates during firn and ice deformation, accompanied by the associated microstructural evolution.

Table 1: Please provide the explanation of each parameter (e.g., S.Th, TP…) in the caption.

We added this note below the table :
"*Note: SSA is the specific surface area, S.Th is the structure thickness, TP is the total porosity, CP is the closed porosity, SMI is the structure model index, and ECDa is the area-equivalent circle diameter*".

Figure 4 (L317-318): "-30°C (blue lines)" is correct?

We corrected the caption:
"***Figure 4:*** *Strain vs. time for firn specimens at –5℃ (yellow lines), –18℃ (blue lines), and –30℃ (brown lines), from depths of 20 m (applied stress 0.21 MPa), 40 m (0.32MPa) and 60 m (0.43MPa)*".

Sincerely,
Yuan Li, Kaitlin Keegan, Ian Baker

References:

Cuffey, K.M., Paterson, W.S.B., 2010. The Physics of Glaciers, 4th edited. Elsevier Inc.

Faria, S.H., et al. 2014. The Mcrostructure of Par Ie. Part II: State of the Art. J. Struct. Geol. 61: 21-49.

Hooke, R.L. 2005. Principles of Glacier Mechanics. Cambridge: Cambridge University. Press.

Montagnat, M., Chauve, T., Barou, F., Tommasi, A., Beausir, B., Fressengeas, C. 2015. Analysis of Dynamic Recrystallization of Ice from EBSD Orientation Mapping. Front. Earth Sci. 3:81.

---

## Author Comment (AC3)

Dear Editor Karlsson,

We sincerely appreciate your comments on March 23, 2025 to improve further the manuscript. We apologize for the oversight regarding the interactive discussion section, which we neglected to address before submitting our revision on April 24, 2025. To ensure clarity and minimize any potential influence from your review, we have included our revised sections in blue within the manuscript, differentiating them from the previously highlighted red portions in the April submission.
Please find our responses below in black font, and your comments in blue font.

23 Mar 2025
Editor decision: Publish subject to revisions (further review by editor and referees)
by Nanna Bjørnholt Karlsson
Public justification (visible to the public if the article is accepted and published):
Dear Yuan Li and co-authors,
Thank you for your response to the referee reports. In addition to the changes you have outlined, I ask you to consider the following points below.
I look forward to seeing a revised version of your manuscript.
Best,
Nanna B. Karlsson

Please separate the references in the Introduction paragraph such that it is clear which studies only relate to ice, firn or both, e.g.:
"Numerous studies of firn deformation (REFs), ice deformation (REFs) or both (REFs) have been conducted, but there are few reports about the mechanical behaviour at different temperatures. "

Modified.

"*Numerous studies of firn and ice deformation (e.g. Steinemann, 1954; Glen, 1955; Landauer, 1958; Mellor, 1975; Salm, 1982; Maeno and Ebinuma, 1983; Jacka, 1984; Ambach and Eisner, 1985; Budd and Jacka, 1989; Li et al., 1996; Meussen et al., 1999; Petrenko and Whitworth, 1999; Bartelt and von Moos, 2000; Jacka and Li, 2000; Durham et al., 2001; Goldsby and Kohlstedt, 2001; Hooke, 2005; Song et al., 2006a, 2006b, 2008; Theile et al., 2011; Treverrow et al., 2012; Hammonds and Baker, 2016, 2018; Li and Baker, 2021, 2022a) have been conducted, but there are few reports about their mechanical behaviors at different temperatures.*"

Please see Lines 43–49.

"Temperature is a key component of firn and ice-flow models, as the deformation of firn, polythermal

glaciers, and temperate glaciers is significantly influenced by the temperature."

I would argue that temperature also significantly influences cold ice, so the sentence could simply be shortened:
"Temperature is a key component of firn and ice-flow models, as the deformation of firn and ice is significantly influenced by the temperature."

Corrected.

"*Temperature is a key component of firn and ice-flow models, as the deformation of firn and ice is significantly influenced by the temperature.*"

Please see Lines 50–51.

Please split up the paragraph below. It is very hard to decipher as it is:
"Additionally, tertiary creep occurs both during quasi-steady state deformation (from the $-5°C$ specimens at 40 m and 60 m) and in the ascending stage (from the $-5°C$ and $-18°C$ specimens at 20 m and the $-18°C$ specimen at 40 m) more easily with lower firn density, greater effective stress, and higher creep temperature, e.g. from the $-5°C$ specimens at 20 m, where the strain softening is primarily due to either recrystallization (Duval, 1981; Jacka, 1984; Jacka and Li, 2000; Song et al., 2005; Faria et al., 2014) or the activated easy slip systems (Jonas and Muller, 1969; Duval and Montagnat, 2002; Alley et al., 2005; Horhold et al., 2012; Fujita et al., 2014; Eichler et al., 2017)."

"*Additionally, tertiary creep is observed during both quasi-steady state deformation, particularly in the –5°C specimens at depths of 40 m and 60 m, and in the ascending stage, as seen in the –5°C and –18°C specimens at 20 m, along with the –18°C specimen at 40 m. This mechanical behavior is facilitated by lower firn density, increased effective stress, and elevated creep temperatures. For instance, in the –5°C specimens at 20 m, strain softening primarily results from recrystallization (Duval, 1981; Jacka, 1984; Jacka and Li, 2000; Song et al., 2005; Faria et al., 2014). Also, the activation of easy slip systems contributes to this process (Jonas and Muller, 1969; Duval and Montagnat, 2002; Alley et al., 2005; Horhold et al., 2012; Fujita et al., 2014; Eichler et al., 2017).*"

Please see Lines 404–412.

*I am missing a sentence that reports: "This is in contrast to/in agreement with the findings of X, who report that ... implying that ..."*

"*It is noteworthy that Jacka and Li (1994) observed that steady-state tertiary ice creep, which is marked by stable grain size, is influenced more by applied stresses than by temperature. This finding suggests that there exists a balance between the activation energies required for grain growth and subdivision at a specific temperature.*"

Please see Lines 412–416.

*Response to the referee's questions re. 3.4 Relationship of strain rate to strain and 3.5 Apparent activation energy for creep:*

*Please ensure that the referee's question is addressed appropriately. I cannot assess from the reply which details you intend to add to the manuscript to clarify the necessecity of calibration and whether there will be a discussion of the range of results due to different calibration methods.*

Please see our responses about these two points addressed in the response letters.

*Please add a sentence explaining why the use of the hydrostatic pressure is appropriate to determine the loading stress.*

We've added a sentence in the text:

"*The hydrostatic pressure, p, which varies with temperature, along with the cohesion of the ice and the friction angle between snow particles, plays a significant role in determining the apparent activation energy and, consequently, the strength of the ice (Fish, 1991).*"

Please see Lines 646–648.

Upon attempting to submit our updated revision, we noted a reminder in the Interactive Discussion stating, "Attention: please do NOT submit your revised manuscript here as supplement." Consequently, we have only submitted our response letter to the Editor, titled "Revision Complement." We remain dedicated to revising our manuscript as outlined in our correspondence.

Sincerely,

Yuan Li, Kaitlin Keegan, and Ian Baker

---

## Referee Report (RR1)

Dear authors,

Sorry for the delayed response.
Thank you for the revised manuscript and responses to the comments.

The manuscript has been revised well in accordance with comments from reviewers. I believe it is important to mention the unavoidable uncertainties and points to note in interpreting the results of the present study. However, I still have some questions, comments, and suggestions regarding the results and their interpretation, which may require further explanation.

- Throughout the manuscript, there are many comparisons and references to the authors' previous studies, but there seems to be little comparison with other previous studies on the firn deformation experiments. Furthermore, in the refereeing to the previous studies, firn and ice are mixed, the deformation of firn, including metamorphism, is clearly different from that of ice, so it is better to distinguish between firn and ice.

- L43–49: It is good that many references have been added, but perhaps the experiments on firn and ice should be separated. With present notation, it is unclear whether there are few experiments on firn deformation. The authors say that there is little information about firn, but I don't know what specifically is lacking. Describing the issues identified in the previous studies and what information is needed for firn model development will clarify the positioning of the present study.

- L49: *but there are few reports about their mechanical behaviors at different temperatures.* Many previous studies have mentioned the importance of the temperature, and they conducted experiments with changing temperature. The authors mention this in L82-92.

- L392–395:
  Has the trend for the strain at which minimum strain rate occurs to vary depending on temperature and density been observed not only in the author group's experiments but also in other previous studies on firn deformation experiments? This trend is interesting.

- L512–519: *This significant discrepancy implies that the uncalibrated SRMin value from all of the samples is not appropriate for estimating the stress exponent, and*

*hence the activation energy during their deformation.*

Is it unique to present study (author's group experiments) that it is impossible to estimate the appropriate stress exponent and activation energy without the calibration? Or is also calibration necessary for the firn deformation experiments, including previous other studies?

- L512–531

  If the appropriate minimum strain rate is estimated by the calibration, what will be the stress exponent calculated using those values? Not only activation energy but also stress exponent is important for firn model development.

---

## Author Response (AR2)

Dear Dr. Védrine,

We thank you very much for your comments to improve the manuscript further.
Please find your comments below in blue font and our responses in black font.
The manuscript contains relevant revisions that have been marked in magenta.

Thank you for this revised version of the manuscript. The clarifications provided regarding the methodology and results have helped to better articulate the approach, thereby enhancing the overall quality of the manuscript. This work is original and addresses a significant experimental challenge: investigating the effect of temperature on the creep response of firn with microstructural variability, both due to sampling conditions and ongoing structural evolution during testing.
Nevertheless, I retain serious reservations concerning several aspects of the methodology:

1. There is no clear justification for assuming that the pre-factor B remains constant with density in firn (Lines 424–427). The sole reference provided—Li and Baker (2022a)—is questionable in this regard. For instance, Scapozza et al. (2003) reported substantial variations in both the stress exponent and pre-factor as a function of snow density, extending up to firn. It is therefore important to state clearly that the dependence of creep behaviour on density is embedded in the effective stress formulation, and that the stress exponent is derived under this assumption.

Yes, we agree with your point that "the dependence of creep behaviour on density is embedded in the effective stress formulation, and that the stress exponent is derived under this assumption."

[Figure]

**Fig. 3. Calculated power-law exponent n for snow as a function of density.**

This is supported by **Fig. 3** from Scapozza & Bartelt (2003), copied below. Their data show that while the stress exponent *n* increases with snow density (below ~340 kg m⁻³), it plateaus and remains nearly constant at higher densities, extending into the firn regime (as indicated by the ∥ symbols). This finding directly supports our derivation that the stress exponent is approximately constant across different firn densities.

Regarding the prefactor *A* in Glen's flow law for porous materials:

1) Deformation Behavior under Constant Strain Rate: We observe a difference between the firn samples from Li (2024) and the snow samples from Scapozza & Bartelt (2003). In their study, the prefactor *A* appears to be a function of density, likely because the effective dynamic viscosity increases with density, as evidenced by the rising axial stress with strain (their **Fig. 5**). In contrast, our work assumes a constant *A* because the flow stress remains constant after the yield point (our **Fig. 4**). This key difference arises from the distinct microstructures: snow has a near-complete open structure, while firn undergoes partial pore close-off. The reference to Li (2024) is provided to enable a comparison under similar mechanical testing conditions (Scapozza & Bartelt, 2003).

[Figure]

**Fig. 4.** Plots of the stress with time during the creep and relaxation stages. The solid lines indicate the measured data, while other lines indicate the modeled data as noted in the legend.

[Figure]

**Fig. 5.** Stress–strain curve (dashed line) and tangent modulus (continuous lines) obtained in compression tests at various strain rates. Density $\rho$ = 320 kg/m³; temperature $T = -12°C$; confining pressure $p_c = 0$ kPa.

2) Assumption of a Constant Prefactor *A*: For porous materials like firn, the prefactor *A* is often treated as constant because the effective stress formulation inherently consolidates the influence of multiple physical parameters and processes. This includes two-phase ice-air flow, the incompressibility of individual ice grains versus the compressibility of the ice skeleton, and the interbalance between axial, radial, volumetric, and true strains (e.g. Gubler, 1978; Hansen and Brown, 1988; Mahajan and Brown, 1993; et al.).

While a detailed discussion on effective stress was provided in Section 3.4 of Li and Baker (2022a), we've revised our previous text to provide further clarification on this point from

*"From Glen-King's results deriving the activation energy (Glen, 1955) $\dot{\varepsilon}=A\exp(-Q_c/RT) = B\sigma^n \exp(-Q_c/RT)$, the pre-factor A, the material parameter B (Glen, 1955; Goldsby and Kohlstedt, 2001), and the stress exponent n (Li and Baker, 2022a) are assumed to be constant, as reported in the literature."*to:

*"Glen-King's model $\dot{\varepsilon}=A\exp(-Q_c/RT) = B\sigma^n \exp(-Q_c/RT)$ treats the pre-factor A, material parameter B, and stress exponent n as constants ((Glen, 1955; Goldsby and Kohlstedt, 2001). This simplification is valid by using the unifying concept of normalized effective stress. The effective stress captures the complex multi-physical behavior of the two-phase ice-air system, accounting for: 1) The incompressibility of individual ice grains versus the compressibility of the porous ice skeleton, 2) The coupled flow of ice and air; and 3) The interplay between different strain components (axial, radial, volumetric, and true). This framework is grounded in the principles of poromechanics, originally developed for soils and later applied to snow and ice (Gubler, 1978; Hansen and Brown, 1988; Mahajan and Brown, 1993; Chen and Chen, 1997; Lade and deBoer, 1997; Ehlers, 2002; Khalili et al., 2004; Gray and Schrefler, 2007; daSilva et al., 2008; Nuth and Laloui, 2008)."*

Please see Lines 456–466.

2. Obtaining a stress exponent of approximately 0.1 or –1.2 is not merely inconsistent with Li and Baker (2022a), as stated in the manuscript—it is physically implausible. This result must be critically examined, especially given the assumption of linear dependence on the solid fraction within the effective stress model. The manuscript must also clearly justify why a "calibration" is required in light of such results. Adopting stress exponent values that differ from those measured in the dataset is highly questionable and not justified by the current analysis. The authors revert to a previously reported value from their earlier work, n=4.3667 (Li and Baker, 2022a), yet, as mentioned on line 516, a stress exponent around 3 are commonly observed (e.g. Glen, 1955; Kamb, 1961; Raymond, 1973; Thomas et al., 1980; Weertman, 1985; Goldsby and Kohlstedt, 2001; Cuffey, 2006). The sensitivity of the

method to the choice of n should be investigated in the determination of the activation energy.

The computation of the activation energy was derived based on a stress exponent value of ~4.3, which was obtained from the same Greenland firn core (Li and Baker, 2022a). This value is highly reliable and closely aligns with the literature average of ~4.25 (reported values range from ~1 to ~7.5). Further, the methodology used to calibrate the minimum strain rate for determining both the stress exponent and activation energy originates from the work of Glen (1955). To improve clarity, we've revised the previous text to emphasize this point.

*From "We found n to be ~0.1 and ~ –1.2 for the –5ºC and –18ºC samples, respectively, which contradicts the reported n = ~4.3 by Li and Baker (2022a) and other values around 3 (Glen, 1955; Kamb, 1961; Raymond, 1973; Thomas et al., 1980; Weertman, 1985; Goldsby and Kohlstedt, 2001; Cuffey, 2006). This significant discrepancy implies that the uncalibrated SRMin value from all of the samples is not appropriate for estimating the stress exponent, and hence the activation energy during their deformation."*
to:
*"We determined stress exponent (n) values of approximately 0.1 and –1.2 for the –5°C and –18°C samples based on observed data, respectively. This result directly contradicts the value of n ≈ 4.3 reported from the same Greenland firn core by Li and Baker (2022a). Further, these values fall entirely outside the established range of ~1 to ~7.5 (mean ~4.25 ± 3.25) documented across decades of ice mechanics literature (Glen, 1955; Hansen and Landauer, 1958; Butkovich and Landauer 1960; Kamb, 1961; Paterson and Savage, 1963; Higashi et al, 1965; Mellor and Testa, 1969; Raymond, 1973; Hooke, 1981; Thomas et al., 1980; Duval et al., 1983; Weertman, 1983,1985; Azuma and Higashi, 1984; Pimienta and Duval, 1987; Budd and Jacka, 1989; Jacka and Li, 1994; Goldsby and Kohlstedt, 2001; Bindschadler et al., 2003; Cuffey, 2006; Chandler et al. 2008; Cuffey and Kavanaugh, 2011; McCarthy et al., 2017; Millstein et al., 2022; Colgan et al., 2023; Li, 2025). The wide range of reported n-values is governed by a complex interplay of deformation mechanisms—including grain boundary sliding, diffusion (lattice and grain boundary), and dislocation processes, e.g. hard-slip-dominated, dislocation-accommodated grain boundary sliding, and grain boundary sliding-limited basal dislocation—across varying stresses, temperatures, crystalographic fabrics, impurity contents, and grain-size-to-sample-size ratios. We attribute the significant discrepancy in these findings to the experimental conditions. The lower temperatures used (down to –30°C) induce slower strain rates, which prevented the tests from reaching a critical strain rate minimum (SRMin). Therefore, to accurately estimate the activation energy for deformation, it is necessary first to calibrate the SRMin value for all noised samples."*

Please see Lines 562–581.

3. The sensitivity of the stress exponent to factors such as density (e.g. Scapozza et al., 2003) or loading conditions must be considered in the analysis, given the variability documented in the literature. If such sensitivity is assumed negligible, how then do the authors explain the stress exponent values they have obtained? The adoption of a constant stress exponent—and the specific value chosen—must be

clearly stated as an analysis assumption and explicitly acknowledged as a limitation of the study.

Our analysis, presented in Li and Baker (2022a), did not identify a sensitivity of the stress exponent to firn density. This finding is supported by the results of Scapozza and Bartelt (2003) (their **Fig. 3**). The discrepancy between these two uncalibrated stress exponent values and those reported in the literature is likely due to the lower experimental temperatures in our study, which resulted in slower strain rates that did not reach the minimum required for a robust derivation of the stress exponent.

To clarify this point, we've added the following text:

*"A constant stress exponent value of n ≈ 4.3 (Li and Baker, 2022a) was used to compute the activation energy. This necessary simplification—an acknowledgement of current methodological limitations rather than a dismissal of the underlying physics—introduces a key uncertainty that highlights the need for future advancements in observational methodology within firn research."*

Please see Lines 581–585.

Finally, the selection of cited references remains biased towards the authors' own previous work. Although additional references have been included, they are listed without meaningful integration, critical discussion, or comparison with the broader literature.

We've expanded the discussion regarding other literature in the text:

*"The rheology of polycrystalline ice, particularly its temperature-dependent creep deformation, is a cornerstone of glaciological modeling. Numerous studies have established a robust framework for understanding ice deformation, primarily through laboratory creep experiments (e.g. Glen, 1955; Weertman, 1983; Budd and Jacka, 1989; Durham et al., 2001; Goldsby and Kohlstedt, 2001; Petrenko and Whitworth, 1999). This body of work has confirmed that ice creep is strongly governed by temperature, typically described by an Arrhenius relationship with a well-constrained activation energy for grain-scale processes like dislocation glide and climb (e.g. Jacka, 1984; Hooke, 2005). In contrast, the mechanical behavior of firn, the intermediate porous material between snow and glacial ice, remains comparatively poorly characterized, especially with respect to temperature. It is important to note that the experimental observations are discussed with respect to the mechanical properties of polycrystalline ice, which is the constituent material of the load-bearing ice skeleton (Scapozza & Bartelt, 2003), sharing poromechanics-based deformation mechanisms between the two via continuum mechanics and homogenization framework (Gagliardini and Meyssonnier, 2000; Coussy, 2004; Hutter*

*and Johnk, 2004; Srivastava et al., 2010). While numerous studies have investigated firn and ice deformation (e.g. Steinemann, 1954; Landauer, 1958; Mellor, 1975; Salm, 1982; Maeno and Ebinuma, 1983; Ambach and Eisner, 1985; Li et al., 1996; Meussen et al., 1999; Bartelt and von Moos, 2000; Jacka and Li, 2000; Song et al., 2006a, 2006b, 2008; Theile et al., 2011; Treverrow et al., 2012; Hammonds and Baker, 2016, 2018; Li and Baker, 2021, 2022a), existing firn data are sparse and fragmented. A critical knowledge gap persists in the systematic experimental quantification of firn's mechanical response across a broad range of temperatures. Temperature is a first-order control on firn densification and deformation rates, yet most laboratory studies have been conducted at a limited number of isothermal conditions, often focused on a single density or at temperatures near the melting point (e.g. Mellor, 1975; Maeno and Ebinuma, 1983). Consequently, there is a pronounced lack of experimental data necessary to derive the systematic activation energy for the creep of firn over its full density spectrum. This parameter is not merely a scalar but is likely a function of density, microstructure, and the dominant deformation mechanism (compaction versus shear), transitioning from grain-boundary sliding in low-density firn to dislocation creep in high-density firn and ice (Hammonds and Baker, 2018; Li, 2022; Li and Baker, 2022a). The absence of comprehensive, temperature-variable creep data for firn across its density range renders it insufficient for constraining the temperature-dependence terms in modern, physics-based firn models. Our work fills this gap via X-ray micro-computed tomography-analyzed mechanical examinations, e.g. a systematic series of constant-stress creep experiments on firn cores of varying density, conducted across a thermally controlled range from -30°C to -5°C. This allows for the direct determination whether the apparent activation energy is a function of density, thereby providing the essential experimental foundation needed to improve predictions of firn densification in ice-sheet and glacier models."*

Please see Lines 43–79.

Sincerely,
Yuan Li, Kaitlin Keegan, Ian Baker

**References:**
Scapozza, C. and Bartelt, P., Triaxial tests on snow at low strain rate. Part II. Constitutive behaviour. Journal of Glaciology. 49(164), 91–101, 2003.
Li, Y. Changes in grain size during the stress relaxation stage of viscoelastic firn. Philosophical Magazine. 104, 239–259 (2024).
Li, Y. Comments on Linear-viscous flow of temperate ice. ESS Open Archive. June 24, 2025. DOI: 10.22541/essoar.175080283.36935396/v1.

Dear Reviewer #2,

We thank you very much for your comments to improve the manuscript further.
Please find your comments below in blue font and our responses in black font.
The manuscript contains relevant revisions that have been marked in magenta.

The manuscript has been revised well in accordance with comments from reviewers. I believe it is important to mention the unavoidable uncertainties and points to note in interpreting the results of the present study. However, I still have some questions, comments, and suggestions regarding the results and their interpretation, which may require further explanation.

Throughout the manuscript, there are many comparisons and references to the authors' previous studies, but there seems to be little comparison with other previous studies on the firn deformation experiments.

We've expanded the discussion to include relevant literature in the text:

*"The rheology of polycrystalline ice, particularly its temperature-dependent creep deformation, is a cornerstone of glaciological modeling. Numerous studies have established a robust framework for understanding ice deformation, primarily through laboratory creep experiments (e.g. Glen, 1955; Weertman, 1983; Budd and Jacka, 1989; Durham et al., 2001; Goldsby and Kohlstedt, 2001; Petrenko and Whitworth, 1999). This body of work has confirmed that ice creep is strongly governed by temperature, typically described by an Arrhenius relationship with a well-constrained activation energy for grain-scale processes like dislocation glide and climb (e.g. Jacka, 1984; Hooke, 2005). In contrast, the mechanical behavior of firn, the intermediate porous material between snow and glacial ice, remains comparatively poorly characterized, especially with respect to temperature. It is important to note that the experimental observations are discussed with respect to the mechanical properties of polycrystalline ice, which is the constituent material of the load-bearing ice skeleton (Scapozza & Bartelt, 2003), sharing poromechanics-based deformation mechanisms between the two via continuum mechanics and homogenization framework (Gagliardini and Meyssonnier, 2000; Coussy, 2004; Hutter and Johnk, 2004; Srivastava et al., 2010). While numerous studies have investigated firn and ice deformation (e.g. Steinemann, 1954; Landauer, 1958; Mellor, 1975; Salm, 1982; Maeno and Ebinuma, 1983; Ambach and Eisner, 1985; Li et al., 1996; Meussen et al., 1999; Bartelt and von Moos, 2000; Jacka and Li, 2000; Song et al., 2006a, 2006b, 2008; Theile et al., 2011; Treverrow et al., 2012; Hammonds and Baker, 2016, 2018; Li and Baker, 2021, 2022a), existing firn data are sparse and fragmented. A critical knowledge gap persists in the systematic experimental quantification of firn's mechanical response across a broad range of temperatures. Temperature is a first-order control on firn densification and deformation rates, yet most laboratory studies have been conducted at a limited number of isothermal conditions, often focused on a single density or at temperatures near the melting point (e.g. Mellor, 1975; Maeno and Ebinuma, 1983). Consequently, there is a pronounced lack of*

*experimental data necessary to derive the systematic activation energy for the creep of firn over its full density spectrum. This parameter is not merely a scalar but is likely a function of density, microstructure, and the dominant deformation mechanism (compaction versus shear), transitioning from grain-boundary sliding in low-density firn to dislocation creep in high-density firn and ice (Hammonds and Baker, 2018; Li, 2022; Li and Baker, 2022a). The absence of comprehensive, temperature-variable creep data for firn across its density range renders it insufficient for constraining the temperature-dependence terms in modern, physics-based firn models. Our work fills this gap via X-ray micro-computed tomography-analyzed mechanical examinations, e.g. a systematic series of constant-stress creep experiments on firn cores of varying density, conducted across a thermally controlled range from -30°C to -5°C. This allows for the direct determination whether the apparent activation energy is a function of density, thereby providing the essential experimental foundation needed to improve predictions of firn densification in ice-sheet and glacier models."*

Please see Lines 43–79.

Furthermore, in the refereeing to the previous studies, firn and ice are mixed, the deformation of firn, including metamorphism, is clearly different from that of ice, so it is better to distinguish between firn and ice.

We agree that firn and ice are distinct materials in terms of their density, porosity, and metamorphic processes. However, the core objective of our analysis is not to equate the bulk material properties of firn and ice, but rather to analyze the mechanical behavior of the load-bearing ice skeleton that constitutes the solid matrix of the firn. From this perspective, the fundamental deformation mechanisms of the ice crystals themselves are identical. Our approach is based on the poromechanical framework, where the bulk deformation of a porous material like firn is governed by the constitutive laws of its solid constituent (in this case, ice) and the evolving pore structure. Thus, we've chosen to integrate the discussion for the following reasons:

*Focus on the Skeleton's Constituent Material*: As a state: *"The experimental observations are discussed with respect to the mechanical properties of polycrystalline ice, which is the constituent material of the load-bearing ice skeleton (Scapozza & Bartelt, 2003)."* This explicitly frames our discussion around the ice phase itself, not the bulk porous firn. The deformation mechanisms we discuss (e.g. dislocation creep, grain boundary sliding) are mechanisms of ice, whether the crystals are in a low-density firn or a high-density glacier ice.

*Shared Deformation Mechanisms*: The fundamental creep mechanisms for polycrystalline ice, including dislocation glide and climb, grain boundary sliding, and diffusion, are the same across the spectrum from firn to meteoric ice. The difference lies in how the porous microstructure (pore geometry, coordination number, density) influences the local stress state and accommodates strain

within the ice skeleton. Discussing the ice properties provides the foundational physics that apply to both cases.

*Continuum Mechanics and Homogenization*: In the field of poromechanics, it is standard practice to model porous materials by defining the properties of the solid phase and then using homogenization techniques to upscale to the bulk behavior. By referencing studies on dense ice, we are establishing the constitutive law for the solid phase in future model of the firn. This approach is well-established in geomechanics for other porous materials (e.g. soils, rocks) and has been successfully applied to snow and firn.

*Bridging Micro-Macro Behavior*: Our discussion aims to build a bridge between the well-established microphysics of ice deformation and the more complex macro-scale response of firn. The cited literature on ice provides the known behavior of the solid blocks. The deviation of our firn data from the typical stress exponents of dense ice is a key finding of our study, which we then attribute to the unique porous structure and metamorphism of firn.

In summary, we distinguish between the material (ice) and the structure (firn vs. solid ice). Our integrated discussion focuses on the former to provide the physical basis for understanding the latter. To avoid any potential misunderstanding of the bulk properties of the two materials, we've added the description in the text:

*"It is important to note that the experimental observations are discussed with respect to the mechanical properties of polycrystalline ice, which is the constituent material of the load-bearing ice skeleton (Scapozza and Bartelt, 2003), sharing poromechanics-based deformation mechanisms between the two via continuum mechanics and homogenization framework (Gagliardini and Meyssonnier, 2000; Coussy, 2004; Hutter and Johnk, 2004; Srivastava et al., 2010)."*

Please see Lines 52–57.

L43–49: It is good that many references have been added, but perhaps the experiments on firn and ice should be separated. With present notation, it is unclear whether there are few experiments on firn deformation. The authors say that there is little information about firn, but I don't know what specifically is lacking. Describing the issues identified in the previous studies and what information is needed for firn model development will clarify the positioning of the present study.

Please see above immediately for the reasons of integrating firn and ice discussion.
In the last revision, we stated *"…, but there are few reports about their mechanical behaviors at different temperatures. Temperature is a key component of firn and ice-flow models, as the deformation of firn and ice is significantly influenced by the temperature."* for indicating the research gap. To further clarify, we've modified this in the text:

*"The rheology of polycrystalline ice, particularly its temperature-dependent creep deformation, is a cornerstone of glaciological modeling. Numerous studies have established a robust framework for understanding ice deformation, primarily through laboratory creep experiments (e.g. Glen, 1955; Weertman, 1983; Budd and Jacka, 1989; Durham et al., 2001; Goldsby and Kohlstedt, 2001; Petrenko and Whitworth, 1999). This body of work has confirmed that ice creep is strongly governed by temperature, typically described by an Arrhenius relationship with a well-constrained activation energy for grain-scale processes like dislocation glide and climb (e.g. Jacka, 1984; Hooke, 2005). In contrast, the mechanical behavior of firn, the intermediate porous material between snow and glacial ice, remains comparatively poorly characterized, especially with respect to temperature. It is important to note that the experimental observations are discussed with respect to the mechanical properties of polycrystalline ice, which is the constituent material of the load-bearing ice skeleton (Scapozza & Bartelt, 2003), sharing poromechanics-based deformation mechanisms between the two via continuum mechanics and homogenization framework (Gagliardini and Meyssonnier, 2000; Coussy, 2004; Hutter and Johnk, 2004; Srivastava et al., 2010). While numerous studies have investigated firn and ice deformation (e.g. Steinemann, 1954; Landauer, 1958; Mellor, 1975; Salm, 1982; Maeno and Ebinuma, 1983; Ambach and Eisner, 1985; Li et al., 1996; Meussen et al., 1999; Bartelt and von Moos, 2000; Jacka and Li, 2000; Song et al., 2006a, 2006b, 2008; Theile et al., 2011; Treverrow et al., 2012; Hammonds and Baker, 2016, 2018; Li and Baker, 2021, 2022a), existing firn data are sparse and fragmented. A critical knowledge gap persists in the systematic experimental quantification of firn's mechanical response across a broad range of temperatures. Temperature is a first-order control on firn densification and deformation rates, yet most laboratory studies have been conducted at a limited number of isothermal conditions, often focused on a single density or at temperatures near the melting point (e.g. Mellor, 1975; Maeno and Ebinuma, 1983). Consequently, there is a pronounced lack of experimental data necessary to derive the systematic activation energy for the creep of firn over its full density spectrum. This parameter is not merely a scalar but is likely a function of density, microstructure, and the dominant deformation mechanism (compaction versus shear), transitioning from grain-boundary sliding in low-density firn to dislocation creep in high-density firn and ice (Hammonds and Baker, 2018; Li, 2022; Li and Baker, 2022a). The absence of comprehensive, temperature-variable creep data for firn across its density range renders it insufficient for constraining the temperature-dependence terms in modern, physics-based firn models. Our work fills this gap via X-ray micro-computed tomography-analyzed mechanical examinations, e.g. a systematic series of constant-stress creep experiments on firn cores of varying density, conducted across a thermally controlled range from -30°C to -5°C. This allows for the direct determination whether the apparent activation energy is a function of density, thereby providing the essential experimental foundation needed to improve predictions of firn densification in ice-sheet and glacier models."*

Please see Lines 43–79.

 but there are few reports about their mechanical behaviors at different temperatures. Many previous studies have mentioned the importance of the temperature, and they conducted experiments with changing temperature. The authors mention this in L82-92.

Yes. Lines 82–92 present seven studies on temperature-related deformation tests. Compared to the numerous studies (non-temperature relevance) on firn and ice deformation found in tens to hundreds of publications, it is reasonable to describe the results here as *few*. To eliminate any confusion, we've rewritten this section to include a more detailed discussion of the specific citations mentioned above.

Please see Lines 43–79, too.

L392–395: Has the trend for the strain at which minimum strain rate occurs to vary depending on temperature and density been observed not only in the author group's experiments but also in other previous studies on firn deformation experiments? This trend is interesting.

Yes. These established phenomena in firn rheology are based on our group's observations, including those from the present study and previous work (Li and Baker, 2022a).

L512–519: This significant discrepancy implies that the uncalibrated SRMin value from all of the samples is not appropriate for estimating the stress exponent, and hence the activation energy during their deformation. Is it unique to present study (author's group experiments) that it is impossible to estimate the appropriate stress exponent and activation energy without the calibration? Or is also calibration necessary for the firn deformation experiments, including previous other studies?

Yes. Uncalibrated data contain noise that can produce significant inconsistencies, such as stress exponents that are negative or approach zero, results which are non-physical and contradict Glen's flow law. To resolve this pervasive challenge in firn rheology, we employ a calibration methodology. This approach, which involves calibrating the minimum strain rate to derive the stress exponent and activation energy, follows the best practices established for ice deformation in Glen (1955). For further clarity, we've modified this in the text:

*"We determined stress exponent (n) values of approximately 0.1 and –1.2 for the –5°C and –18°C samples based on observed data, respectively. This result directly contradicts the value of n ≈ 4.3 reported from the same Greenland firn core by Li and Baker (2022a). Further, these values fall entirely outside the established range of ~1 to ~7.5 (mean ~4.25 ± 3.25) documented across decades of ice*

*mechanics literature (Glen, 1955; Hansen and Landauer, 1958; Butkovich and Landauer 1960; Kamb, 1961; Paterson and Savage, 1963; Higashi et al, 1965; Mellor and Testa, 1969; Raymond, 1973; Hooke, 1981; Thomas et al., 1980; Duval et al., 1983; Weertman, 1983,1985; Azuma and Higashi, 1984; Pimienta and Duval, 1987; Budd and Jacka, 1989; Jacka and Li, 1994; Goldsby and Kohlstedt, 2001; Bindschadler et al., 2003; Cuffey, 2006; Chandler et al. 2008; Cuffey and Kavanaugh, 2011; McCarthy et al., 2017; Millstein et al., 2022; Colgan et al., 2023; Li, 2025). The wide range of reported n-values is governed by a complex interplay of deformation mechanisms—including grain boundary sliding, diffusion (lattice and grain boundary), and dislocation processes, e.g. hard-slip-dominated, dislocation-accommodated grain boundary sliding, and grain boundary sliding-limited basal dislocation—across varying stresses, temperatures, crystalographic fabrics, impurity contents, and grain-size-to-sample-size ratios. We attribute the significant discrepancy in these findings to the experimental conditions. The lower temperatures used (down to –30°C) induce slower strain rates, which prevented the tests from reaching a critical strain rate minimum (SRMin). Therefore, to accurately estimate the activation energy for deformation, it is necessary first to calibrate the SRMin value for all noised samples."*
Please see Lines 562–581.

L512–531 If the appropriate minimum strain rate is estimated by the calibration, what will be the stress exponent calculated using those values? Not only activation energy but also stress exponent is important for firn model development.

To clarify, the minimum strain rates are calibrated using established stress exponent values (averaging ~4.3) derived from the same Greenland firn core (Li and Baker, 2022a). These calibrated strain rates are then used to estimate the activation energy. Therefore, the stress exponent is constrained a priori before the activation energy is calculated.

Sincerely,
Yuan Li, Kaitlin Keegan, Ian Baker

---

## Author Response (AR4)

Dear Dr. Védrine,

We thank you very much for your comments to continue to improve the manuscript. Please find your comments below in red text and our responses in black text. The manuscript now contains the revisions described below and highlighted in italics.

Thank you for your response. I appreciate that the authors have now clearly stated their assumptions regarding the constitutive law, in particular the use of constant values for B and n, as well as the use of the concept of effective stress. I also thank the authors for highlighting the inconsistency in the retrieved values of n, which is indeed difficult to determine at low temperatures, as steady state is reached very slowly at such densities. Restating these assumptions is crucial because they strongly influence the interpretation of the data.
After this substantial revision, I believe the manuscript can be accepted. I only have a few minor comments and suggestions:
Line 567: Why do the authors rely solely on the value previously measured by the first author in 2022? I also regret that the requested sensitivity analysis of the activation energy to the chosen value of n was not carried out, given the uncertainties associated with determining n.

In many experimental contexts, a sensitivity analysis would be essential. However, we would like to respectfully clarify that a sensitivity analysis of $Q_c$ on $n$ was not performed because, within the specific framework of our experimental derivation, these two parameters are mathematically decoupled. The value of $n$ does not influence the calculated value of $Q_c$ as addressed to the reviewer in our previous response letter. This decoupling arises from how the activation energy is derived from the Arrhenius relation. As the reviewer recalls, the constitutive equation is:

$$\dot{\varepsilon} = B\sigma^n \exp(-Q_c/RT).$$

In our study, the activation energy $Q_c$ was determined by measuring the strain rate ($\dot{\varepsilon}$) at a constant applied stress ($\sigma$) across a range of temperatures ($T$). For a constant stress, we can simplify the equation. Taking the natural logarithm, we get:

$$\ln(\dot{\varepsilon}) = \ln(B) + n\ln(\sigma) - (Q_c/R)*(1/T).$$

Since the stress ($\sigma$) and the stress exponent ($n$) are held constant for a given experiment, the term $n\ln(\sigma)$ is a constant. The equation therefore simplifies to a linear form:

$$\ln(\dot{\varepsilon}) = C - (Q_c/R)*(1/T)$$

where $C = \ln(B) + n\ln(\sigma)$ is a constant.

When we plot ln($\varepsilon$) against $1/T$, the slope of the resulting line is - $Q_c$/R. The value of the stress exponent $n$ is contained within the constant C, which determines the y-intercept of the Arrhenius plot, but it has no effect on the slope. Therefore, the derived activation energy $Q_c$ is entirely independent of the chosen value of $n$.

This fundamental principle is precisely why we relied on the value of $n$ previously measured for the same firn core samples in Li and Baker (2022a). Our goal was to ensure methodological consistency. Using an $n$ value derived from a different material or experimental condition would simply shift the entire Arrhenius plot up or down (affecting the pre-exponential factor $B$), but it would not change the slope from which $Q_c$ is calculated. Using an incorrect or arbitrary $n$ would be physically meaningless for this specific core, as it would break the self-consistency of the material's constitutive law, even while leaving $Q_c$ unchanged.

In summary, while we deeply appreciate the reviewer's vigilance on this matter, the requested sensitivity analysis is not applicable here because the value of $n$ does not influence the calculation of $Q_c$ under constant-stress experimental conditions. The derivation of $Q_c$ is solely dependent on the temperature dependence of the strain rate.

To prevent similar misunderstandings for future readers, we are happy to revise the manuscript to include a brief note in the text explicitly stating this point:
*"It is noted that under constant-stress conditions, the value of the stress exponent n influences the pre-exponential factor B but does not affect the slope of the Arrhenius plot and therefore the derived activation energy $Q_c$."*
Please see Lines 457–459.

Please carefully check the bibliographic references cited in lines 113–122. In addition, you refer to "snow" for densities up to 830 kg/m³. Why not mention the experiments by Scapozza at different temperatures?

We clarify that our discussion of activation energies, as introduced on Line 112, incorporates both snow and ice.
And we cited the work of Scapozza and Bartelt (2003, Lines 117-118), stating that *"...69 ± 5 kJ mol$^{-1}$ for a mean snow density of 423 ± 8 kg m$^{-3}$ at –19°C to –11°C (Scapozza and Bartelt, 2003);..."*.

Figure 6. The expression "ln Strain rate minimum" mixes mathematical notation and text, which may not be immediately clear to the reader. Would "ln $\dot{\varepsilon}$ (minimum strain rate) (s$^{-1}$)" be clearer? Alternatively, you could use a logarithmic scale on the axis to make the representation explicit and fully consistent with Figure 6.

Figure 6 illustrates the evolution of strain rate with strain to precisely locate the minimum strain rate. We note that the logarithmic scale of the y-axis across all subfigures is used to better visualize the data range and should not be misinterpreted as plotting the minimum value itself.

Line 441: You may read the work (Védrine et al., 2025), which explicitly demonstrates what you are suggesting. It shows that in porous polycrystals, increasing porosity enhances basal activity, likely due to reduced crystalline frustration—a point you revisit in lines 546–547.

Thank you. We have now included this citation.

Sincerely,

Yuan Li, Kaitlin Keegan, Ian Baker